# Virulence and Cross-Protection Conferred by an Attenuated Genotype I-Based Chimeric Japanese Encephalitis Virus Strain Harboring the E Protein of Genotype V in Mice

Qiqi Xia,[a] Yan Zhang,[a,b] Yang Yang,[a] Xiaochun Ma,[a] Zhixin Guan,[a] Junjie Zhang,[a] Zongjie Li,[a] Ke Liu,[a] Beibei Li,[a] Donghua Shao,[a] Yafeng Qiu,[a] Jianchao Wei,[a] Zhiyong Ma[a]

aShanghai Veterinary Research Institute, Chinese Academy of Agricultural Sciences, Shanghai, China
bYangtze University, Jingzhou, China

**ABSTRACT** Japanese encephalitis virus (JEV) genotype V (GV) emerged in China in 2009, then South Korea, and has since spread to other regions in Asia and beyond, raising concern about its pathogenicity and the cross-protection offered by JEV vaccines against different genotypes. In this study, we replaced the structural proteins (C-prM-E) of an attenuated genotype I (GI) SD12-F120 strain with those of a virulent GV XZ0934 strain to construct a recombinant chimeric GI-GV JEV (JEV-GI/V) strain to determine the role of the structural proteins in virulence and cross-protection. The recombinant chimeric virus was highly neurovirulent and neuroinvasive in mice. This demonstrated the determinant role of the structural proteins in the virulence of the GV strain. Intracerebral or intraperitoneal inoculation of mice with JEV-GI/V-E5 harboring a combination of substitutions (N47K, L107F, E138K, H123R, and I176R) in E protein, but not mutants containing single substitution of these residues, resulted in decreased or disappeared mortality, suggesting that these residues synergistically, but not individually, played a role in determining the neurovirulence and neuroinvasiveness of the GV strain. Immunization of mice with attenuated strain JEV-GI/V-E5 provided complete protection and induced high neutralizing antibody titers against parental strain JEV-GI/V, but partial cross-protection and low cross-neutralizing antibodies titers against the heterologous GI and GIII strains in mice, suggesting the reduced cross-protection of JEV vaccines among different genotypes. Overall, these findings suggested the essential role of the structural proteins in determination of the virulence of GV strain, and highlighted the need for a novel vaccine against this newly emerged strain.

**IMPORTANCE** The GV JEV showed an increase in epidemic areas, which exhibited higher pathogenicity in mice than the prevalent GI and GIII strains. We replaced a recombinant chimeric GI-GV JEV (JEV-GI/V) strain to determine the role of the structural proteins in virulence and cross-protection. It was found that the essential role of the structural proteins is to determinethe virulence of the GV strain. It is also suggested that there is reduced cross-protection of JEV vaccines among different genotypes, which provides basic data for subsequent JEV prevention, control, and new vaccine development.

**KEYWORDS** Japanese encephalitis virus, genotype V, virulence, vaccine, cross-protection, toxicity

Japanese encephalitis (JE) is an important zoonotic disease caused by Japanese encephalitis virus (JEV) that is mainly transmitted by *Culex* mosquito species among hosts (1). Swine and birds are important in maintaining, amplifying, and spreading the infection, while humans and horses are dead-end hosts (2). JE is mainly prevalent in

Address correspondence to Zhiyong Ma, zhiyongma@shvri.ac.cn, or Jianchao Wei, jianchaowei@shvri.ac.cn.

The authors declare no conflict of interest.

Asia and the Pacific, with more than 3 billion people at risk of infection. Approximately 20%–30% of JE cases are fatal and 30%–50% of survivors have significant neurologic sequelae (3).

JEV is a single-strand positive-sense RNA virus belonging to the *Flaviviridae* family. Its genome is approximately 11 kb in length and contains an open reading frame that encodes a single polyprotein of ~3432 amino acids. Following translation, the polyprotein is proteolytically cleaved into 3 structural proteins (C, prM/M, and E) and 7 nonstructural proteins (NS1, NS2A, NS2B, NS3, NS4A, NS4B, and NS5) (4, 5). The structural proteins play important roles in the viral life cycle, determination of virulence, and induction of an immune response (6–9). In particular, the E protein contributes to receptor-mediated endocytosis and low pH-triggered membrane fusion (10), and is considered to play a major role in the determination of virulence and antigenicity, as well as the elicitation of neutralizing antibodies (11–14).

JEV is divided into 5 genotypes (I to V) according to the *E* gene and genomic sequences (15, 16). Genotype III (GIII) has historically been the main causative agent of JE and was the dominant genotype throughout most of Asia (17). In recent years, genotype I (GI) has displaced GIII as the most frequently isolated genotype in a number of Asian countries including China, Thailand, Korea, Japan, Malaysia, Vietnam, and India (18). Genotype V (GV) (Muar strain) was originally isolated in Singapore in 1952 from a patient who originated in Muar, Malaysia, and that remained the only known occurrence of a GV strain for over 50 years (15, 19). In recent times, GV was detected in *Culex* mosquitoes in China in 2009 (XZ0934 strain) (20) and in South Korea in 2010 and 2012 (21), suggesting that GV is reemerging and spreading into new geographic areas.

Vaccination is the most effective way to control JE infection (17). The vaccines currently licensed, such as the live-attenuated SA14-14-2 vaccine, are derived from a GIII strain (22). Analysis of the cross-protection offered by GIII-derived vaccines demonstrates reduced neutralizing antibody titers against GI strain in children and pigs immunized with GIII-derived vaccines (23, 24), and partial cross-protection against GI strain in mice (25). Similarly, the low protective efficacy of a GIII-derived vaccine against GV strain was observed in mice (26).

Previous observations indicated that GV strain had distinct pathogenicity and antigenicity from GI and GIII strains (27). The antibodies that are raised against GV strain exhibit poor neutralizing activity against GIII strain (28). In addition, the GV strain induces neutralization of antibody production at a slower rate than GIII strain and exhibits higher pathogenicity than the prevalent GI and GIII strains in mice. The structural proteins of flavivirus play important roles in the determination of virulence and the induction of an immune response (6–9). The E protein is associated with virus binding to cellular receptors and membrane fusion, and is responsible for inducing protective immunity (29–31). Mutations at positions 107 or 138 of E protein significantly increase the virulence of the attenuated SA14-14-2 strain (11). The M protein plays important structural roles in the processes of fusion and maturation of progeny virus during cellular infection. It interacts with E protein to form a prM-E heterodimer, which is required for the E protein to fold and form mature virions (32, 33). A single amino acid substitution in the M protein attenuates virulence in mammalian hosts (6). The C protein is indispensable in virus replication and assembly, and is an important target of T-cells during natural infection (9). Therefore, exploring the roles of the structural proteins, especially E protein, in virulence and cross-protection is important for understanding GV pathogenicity, as well as for the development of vaccines against GV strain. In this study, we generated a recombinant chimeric JEV with structural proteins from a virulent GV strain XZ0934 (XZ0934[GV]) using an attenuated GI strain SD12-F120 (SD12-F120[GI]) as the backbone, and examined the roles of structural proteins of the GV strain in virulence and cross-protective efficacy against different genotypes in mice.

## RESULTS

**Construction of chimeric JEV-GI/V harboring structural proteins of virulent GV strain using the attenuated GI strain as a backbone.** The structural proteins of JEV play important roles in the determination of virulence and the induction of an immune

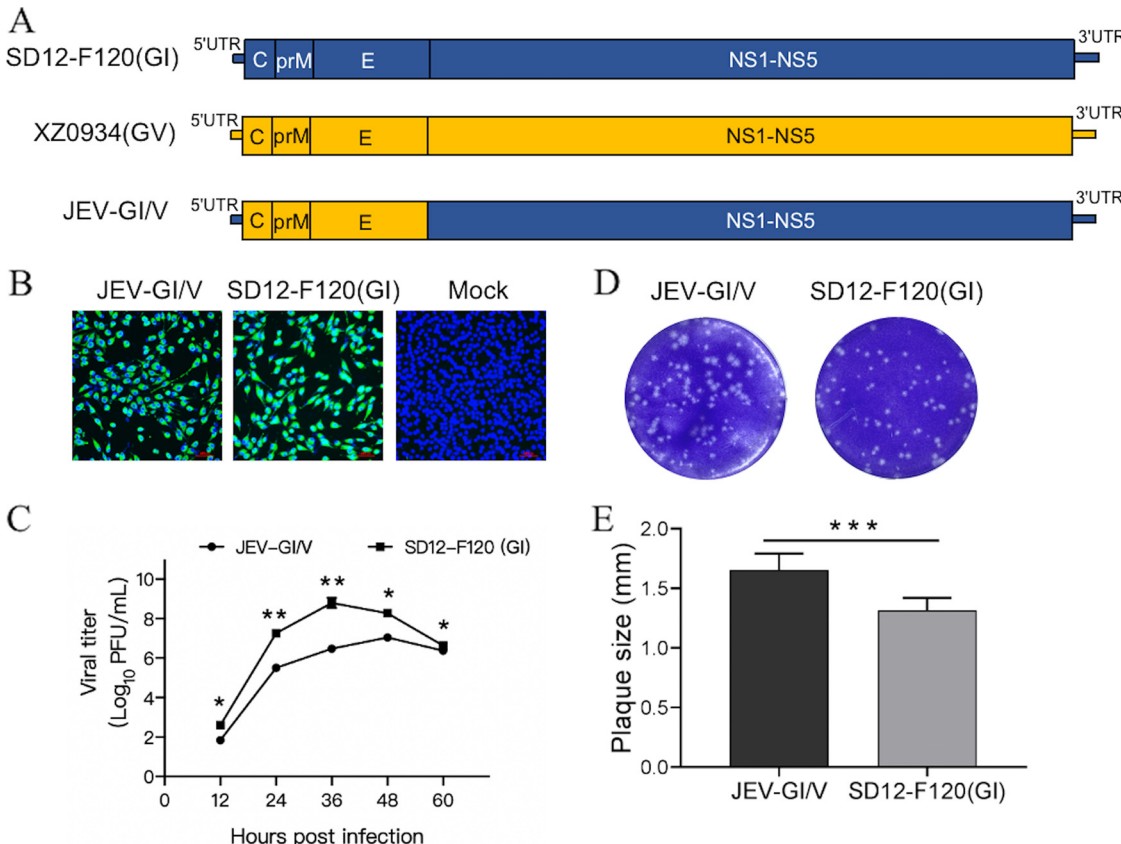

**FIG 1** Construction and characterization of recombinant virus JEV-GI/V. (A) Schematic representation of the chimeric recombinant JEV-GI/V strain. Regions derived from SD12-F120(GI) and XZ0934(GV) strains are shown in different colors. (B) Detection of viral NS3 expression by an immunofluorescence assay. BHK-21 cells were infected with JEV-GI/V and SD12-F120(GI) strains and incubated for 24 h. Expression of viral NS3 was detected by an immunofluorescence assay with anti-NS3 antibodies (green). Nuclei were stained with DAPI (blue). (C) Growth kinetics of recombinant JEV in BHK-21 cells. BHK-21 cells were inoculated with JEV-GI/V or SD12-F120(GI) at 0.01 MOI and the supernatants were collected at the indicated time intervals. JEV titers in the supernatants were measured by $TCID_{50}$ assays. (D) Plaque assays. BHK-21 cells were mock infected or infected with JEV-GI/V or SD12-F120(GI) for analysis of plaque morphology. The plaques were stained with crystal violet at 5 days postinfection. (E) Plaque sizes were measured and plotted. Data are presented as the mean ± SD from 3 independent experiments. A $P$ value was generated by the Student's $t$ test. ***, $P < 0.001$; **, $P < 0.01$; *, $P < 0.05$.

response (6–9). Therefore, the structural proteins (C-prM-E) of the virulent XZ0934(GV) strain (20) were substituted with those in the attenuated SD12-F120(GI) strain to generate an attenuated GI strain-based chimeric JEV harboring the structural proteins of the virulent GV strain (JEV-GI/V) (Fig. 1A). This was then used to determine the role of the GV structural proteins in virulence and cross-protection. The resulting JEV-GI/V strain was verified by DNA sequencing and immunofluorescence assays with anti-NS3 antibodies on BHK-21 cells (Fig. 1B). The replication kinetics of JEV-GI/V were determined by the median tissue culture infective dose ($TCID_{50}$) assays in BHK-21 cells and were compared with its parental SD12-F120(GI) strain. The viral titers of JEV-GI/V in the supernatants peaked at 48 h postinfection (hpi) with a maximum titer of $1.11 \times 10^7$ PFU/mL, while SD12-F120(GI) reached a maximum titer of $6.24 \times 10^8$ PFU/mL at 36 hpi (Fig. 1C). Significant differences in viral titers between the 2 viruses were observed at 12, 24, 36, and 48 hpi. Analysis of the plaque morphology formed on BHK-21 cells indicated that the plaque size of JEV-GI/V was 1.66 mm, which was significantly larger than that of SD12-F120(GI) (1.31 mm) (Fig. 1D and E). This observation demonstrated that chimeric JEV-GI/V replicated less efficiently than its parental SD12-F120(GI) strain in BHK-21 cells.

**Neuroinvasiveness and neurovirulence of JEV-GI/V in mice.** Mice are a well-established small animal model for the evaluation of JEV virulence, including

neuroinvasiveness and neurovirulence (34). We, therefore, inoculated mice intraperitoneally or intracerebrally with JEV-GI/V to test the neuroinvasiveness and neurovirulence, respectively, of this strain. The mice intraperitoneally inoculated with JEV-GI/V developed the clinical signs of JEV infection from a dose of $10^{-2}$ PFU or higher. The median lethal dose ($LD_{50}$) of neuroinvasiveness for JEV-GI/V was $10^{-1.5}$ PFU (Table S1). Additionally, the mice inoculated with JEV-GI/V via the intracerebral route produced the clinical signs of JEV infection, also starting from a dose of $10^{-2}$ PFU. The $LD_{50}$ of neurovirulence for JEV-GI/V was $10^{-2.2}$ PFU (Table S1). These observations suggested that the attenuated phenotype of SD12-F120(GI) was changed by substitution of the structural proteins with those from the virulent XZ0934(GV) strain. Comparison of the virulence of JEV-GI/V with other virulent JEV strains showed that the $LD_{50}$ values of neuroinvasiveness and neurovirulence for JEV-GI/V were less than those of the virulent SD12(GI), N28(GIII), and XZ0934(GV) strains, but likely similar to those of the virulent SA14(GIII) strain (Table S2). Taken together, these data indicated that JEV-GI/V was a virulent strain with high neuroinvasiveness and neurovirulence in mice, suggesting the determinant role of structural proteins in the virulence of GV strains.

**Construction of JEV-GI/V mutants with amino acid substitutions in E protein.** The E protein plays a major role in the determination of JEV virulence (11–14). Therefore, 5 amino acid substitutions, namely, N47K, L107F, E138K, H123R, and I176R, that contribute to the attenuation of JEV virulence (11, 13, 35, 36) were introduced into the E protein of JEV-GI/V, individually or in combination, to generate 6 mutants of JEV-GI/V, which were designated: JEV-GI/V-N47K (N to K substitution at position 47), JEV-GI/V-H107F (H to F substitution at position 107), JEV-GI/V-H123R (H to R substitution at position 123), JEV-GI/V-E138K (E to K substitution at position 138), JEV-GI/V-I176R (I to R substitution at position 176), and JEV-GI/V-E5 (a combination of the N47K, L107F, H123R, E138K, and I176R substitutions) (Fig. 2A). The resulting JEV-GI/V mutants were identified by DNA sequencing and immunofluorescence assays with anti-NS3 antibodies on BHK-21 cells (Fig. 2B). The replication kinetics of these JEV-GI/V mutants were examined by $TCID_{50}$ assays in BHK-21 cells and compared with their parental JEV-GI/V strain. All JEV-GI/V mutants showed viral titers significantly higher than those of JEV-GI/V (Fig. 2C), in particular, the viral titer of JEV-GI/V-E138K peaked at 48 hpi at $2.4 \times 10^8$ PFU/mL, which was remarkably higher than that of JEV-GI/V ($5.01 \times 10^6$ PFU/mL). This observation demonstrated that the introduction of these mutations into E protein led to a significant increase in replication efficiency of JEV-GI/V mutants compared with their parental JEV-GI/V strain in BHK-21 cells. Analysis of the plaque morphology formed on BHK-21 cells indicated that the plaque sizes of JEV-GI/V-N47K, JEV-GI/V-H123R, JEV-GI/V-E138K, and JEV-GI/V-I176R were significantly larger than that of JEV-GI/V, while JEV-GI/V-L107F and JEV-GI/V-E5 formed plaques similar to those of JEV-GI/V with no significant difference in plaque size (Fig. 2D and E).

**Pathogenicity of JEV-GI/V mutants in mice.** Mice were inoculated with JEV-GI/V mutants via the intracerebral route at doses of $10^3$, $10^4$, and $4.5 \times 10^4$ PFU or the intraperitoneal route at doses of $10^3$, $10^4$, and $10^5$ PFU, respectively, and were monitored daily for 21 days. Survival analysis showed that no mice survived in the groups intracerebrally inoculated with JEV-GI/V-N47K, JEV-GI/V-H107F, JEV-GI/V-H123R, JEV-GI/V-E138K, or JEV-GI/V-I176R. However, the mice inoculated with JEV-GI/V-E5 via the intracerebral route showed mortality rates of 0%, 37.5%, and 50% at doses of $10^3$, $10^4$, and $4.5 \times 10^4$ PFU, respectively (Fig. 3A, B, and C, and Table S3). These data suggested that a single mutation at positions 47, 107, 123, 138, and 176 of E protein had no significant effect on the neurovirulence of JEV-GI/V in mice, whereas combined mutations at these positions significantly reduced the neurovirulence of JEV-GI/V in mice.

Survival analysis of mice inoculated via the intraperitoneal route indicated that no mice survived in the groups inoculated with JEV-GI/V-I176R (Table S3), whereas survival was observed in the group inoculated with JEV-GI/V-E138K at a dose of $10^5$ PFU with a mortality rate of 12.5%, as well as in the groups inoculated with JEV-GI/V-N47K, JEV-GI/V-H107F, or JEV-GI/V-H123R at a dose of $10^3$ PFU with a mortality rate of 62.5%. Remarkably, no mortality was observed in groups inoculated with JEV-GI/V-E5 (Fig. 3D,

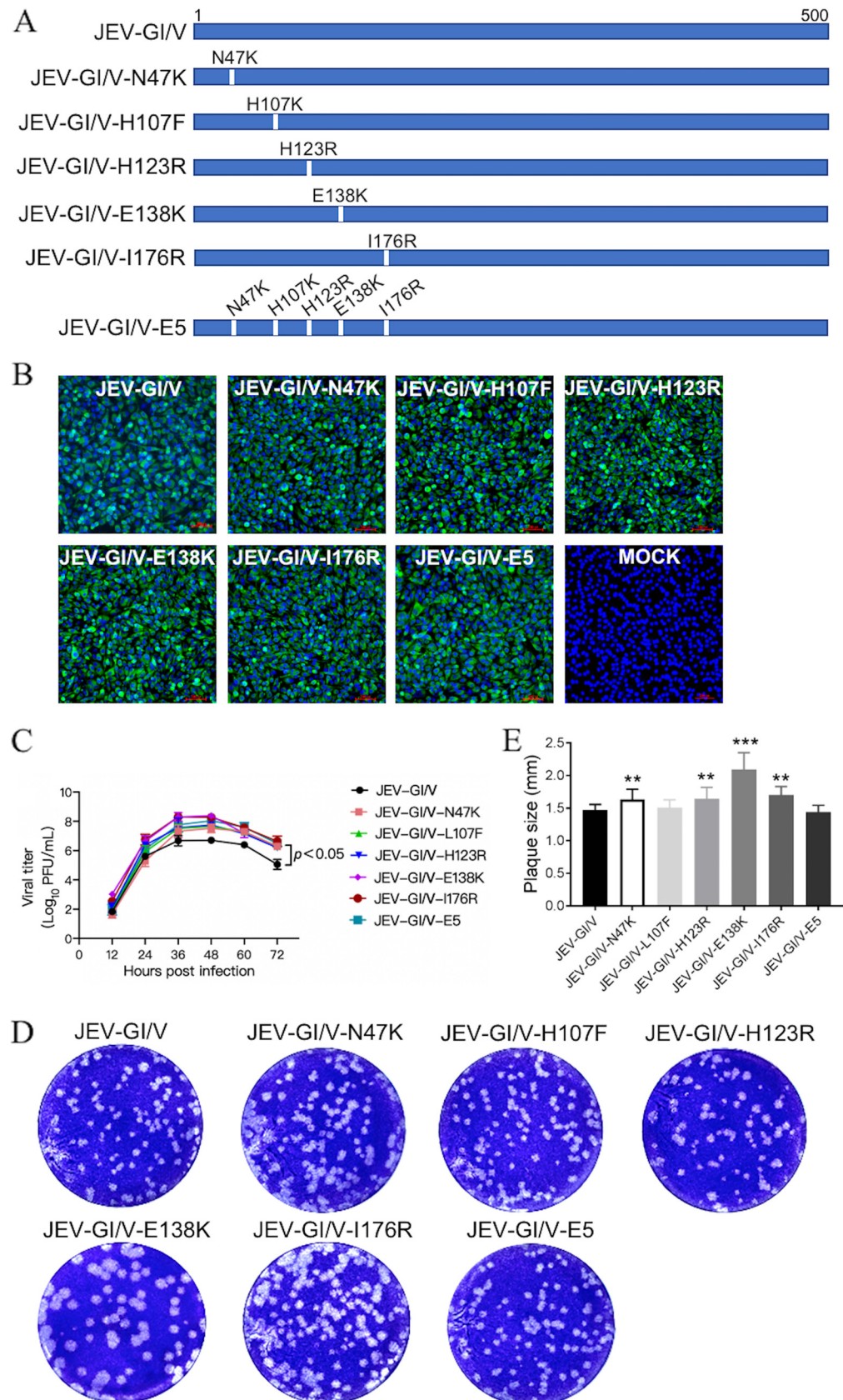

**FIG 2** Construction and characterization of recombinant virus JEV-GI/V mutants. (A) Schematic representation of the recombinant JEV-GI/V mutants. Amino acid substitutions are indicated by white bars. (B) Detection of viral NS3

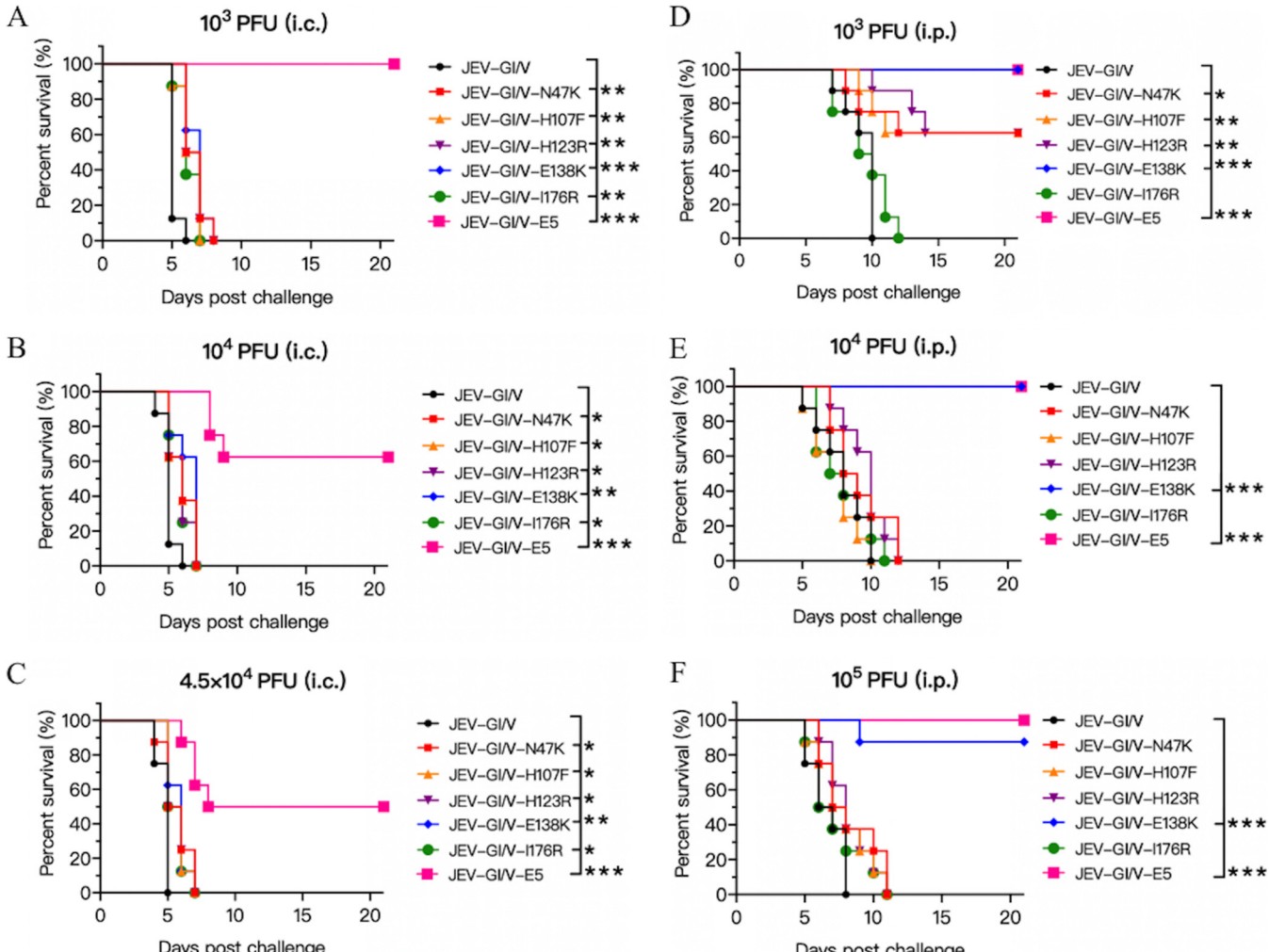

**FIG 3** Survival curves of mice infected with JEV-GI/V mutants. Mice (*n* = 8) were intracerebrally (i.c.) infected with JEV-GI/V and JEV-GI/V mutants at a dose of 10³ PFU (A), 10⁴ PFU (B) and 4.5 × 10⁴ PFU (C). Mice (*n* = 8) were intraperitoneally (i.p.) infected with JEV-GI/V and JEV-GI/V mutants at a dose of 10³ PFU (D), 10⁴ PFU (E) and 10⁵ PFU (F). All mice were monitored for 21 days. The survival rates were calculated and plotted. Survival of each group was statistically compared with JEV-GI/V group using the Kaplan-Meier analysis. ***, $P < 0.001$; **, $P < 0.01$; *, $P < 0.05$.

E, and F, and Table S3). These data suggested that the residues at positions 47, 107, 123, and 138, but not 176, were involved in the neuroinvasiveness of JEV-GI/V, especially the residue at position 138, which played a dominant role in determination of the neuroinvasiveness of JEV-GI/V in mice. Moreover, JEV-GI/V-E5 harboring a combination of N47K, L107F, H123R, E138K, and I176R mutations lost neuroinvasiveness completely at the doses examined, suggesting the synergistic effect of these residues on the determination of neuroinvasiveness, as well as the attenuation of GV virulence in mice.

Viral loads in blood and brain are important for successful invasion of the central nervous system by JEV and other flaviviruses (37, 38). Therefore, blood samples were

**FIG 2** Legend (Continued)
expression by an immunofluorescence assay. BHK-21 cells were infected with the indicated recombinant viruses and incubated for 24 h. Expression of viral NS3 was detected by an immunofluorescence assay with anti-NS3 antibodies (green). Nuclei were stained with DAPI (blue). (C) Growth kinetics of recombinant JEV in BHK-21 cells. BHK-21 cells were inoculated with the indicated viruses at 0.01 MOI and the supernatants were collected at the indicated time intervals. JEV titers in the supernatants were measured by TCID₅₀ assays. Significant differences in virus titers between JEV-GI/V and each JEV-GI/V mutant were tested at 36–72 h postinfection by the Student's *t* test. (D and E) Plaque assays. BHK-21 cells were infected with the indicated viruses for analysis of plaque morphology. The plaques were stained with crystal violet at 5 days (D). The sizes of plaques were measured and plotted (E). Data are presented as the mean ± SD from 3 independent experiments. A *P* value was generated by the Student's *t* test. ***, $P < 0.001$; **, $P < 0.01$; *, $P < 0.05$.

collected from the mice intraperitoneally inoculated with JEV-GI/V mutants at 3 days postinoculation (dpi) and subjected to analysis of viral RNAemia by quantitative real-time reverse transcription PCR (qRT-PCR). Comparison of the RNAemia levels indicated that all JEV-GI/V mutants showed relatively lower levels of RNAemia as compared with JEV-GI/V, but significantly low levels were observed in JEV-GI/V-N47K, JEV-GI/V-E138K, and JEV-GI/V-E5, where, JEV-GI/V-E5 lost completee neuroinvasiveness and displayed the lowest RNAemia level among all groups (Fig. S1A). JEV infection induces inflammatory response in animals. Therefore, the levels of several inflammatory cytokines including interleukin (IL)-6, IL-1$\beta$, tumor necrosis factor (TNF)-$\alpha$, interferon (IFN)-$\beta$ and IFN-$\gamma$ in the blood samples of mice were analyzed (Fig. S1B to F). The mice inoculated with JEV-GI/V produced higher levels of the cytokines analyzed, as compared with the mice inoculated JEV-GI/V mutants. Noticeably, the mice inoculated with JEV-GI/V-E5 that showed completely attenuated phenotype produced the lowest levels of the cytokines among all groups, whereas the mice inoculated with JEV-GI/V-I176K that maintained neuroinvasiveness produced cytokine levels similar to those in mice inoculated with JEV-GI/V (Fig. S1B to F).

To assay viral loads in brains, brain samples were collected the mice intraperitoneally inoculated with JEV-GI/V mutants at 7 dpi and subjected to analysis of viral NS1 gene expression by qRT-PCR. The NS1 gene expression was datable in the mice inoculated with JEV-GI/V, JEV-GI/V-N47K, JEV-GI/V-H107F, JEV-GI/V-H123R, and JEV-GI/V-I176K, with the highest level in JEV-GI/V-inoculated mice (Fig. S2). No expression of NS1 gene was detected in the mice inoculated with JEV-GI/V-E138K or JEV-GI/V-E5 that showed no neuroinvasiveness to mice at the doses of $10^3$ and $10^4$ PFU.

Overall, these observations suggest that a single mutation at positions including 47, 107, 123, and 138 of E protein had no significant effect on the neurovirulence, but attenuated neuroinvasiveness of JEV-GI/V in mice at different levels, as demonstrated by the levels of survivals and viral loads in brains as well as by the levels of viral RNAemia and inflammatory cytokines in blood. Furthermore, the synergistic effect of these residues on determining neurovirulence and neuroinvasiveness of JEV-GI/V was observed in mice.

**Partial protection conferred by the attenuated SD12-F120(GI) strain against JEV-GI/V challenge in mice.** GI virus emerged and became dominant in prevalence in many Asian countries over the past 20 years (18). The live-attenuated GI SD12-F120(GI) strain was previously produced in our laboratory by serial passage of the virulent SD12(GI) strain in BHK-21 cells and showed complete protection against challenge with GI strains in mice (14). We therefore used the SD12-F120(GI) strain as a model GI vaccine to test its cross-protection against JEV-GI/V challenge. Mice were vaccinated with SD12-F120(GI) at high (5,000 PFU), medium (500 PFU), or low (50 PFU) doses, and challenged with homologous genotype SD12(GI) and heterologous genotype JEV-GI/V strains. Vaccination of mice with SD12-F120(GI) completely protected against homologous SD12(GI) strain challenge at the high, medium, and low doses. By contrast, vaccination of mice with SD12-F120(GI) failed to provide complete protection against challenge with the heterologous genotype JEV-GI/V strain. The protection rates against JEV-GI/V challenge were 75% (Fig. 4A), 75% (Fig. 4B), and 50% (Fig. 4C) in groups vaccinated with high, medium, and low doses, respectively. These data indicated that vaccination with the attenuated SD12-F120(GI) strain provided partial protection against heterologous genotype JEV-GI/V challenge.

**Reduced protective efficacy of the SA14-14-2(GIII) vaccine against JEV-GI/V challenge in mice.** The SA14-14-2(GIII) vaccine derived from a GIII strain is currently licensed and is widely used as a JEV vaccine (39). We therefore examined the protective efficacy of SA14-14-2(GIII) vaccine against JEV-GI/V challenge in mice. Mice were immunized with SA14-14-2(GIII) at high (5,000 PFU), medium (500 PFU), or low (50 PFU) doses and challenged with the homologous genotype N28(GIII) and heterologous genotype JEV-GI/V strains. SA14-14-2(GIII) vaccine completely protected the mice following inoculation at high, medium, and low doses against the homologous genotype N28(GIII) strain challenge (Fig. 5). However, the protection rates against heterologous

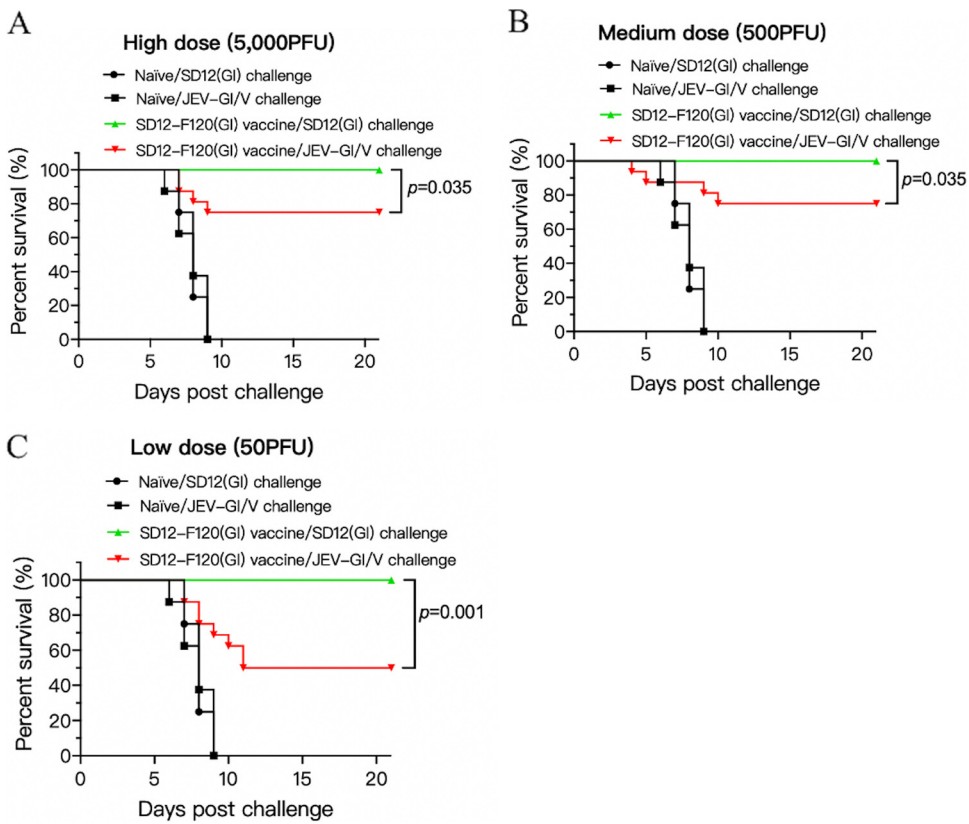

**FIG 4** Protective efficacy of the SD12-F120 (GI) strain. Mice (*n* = 16) were immunized with the attenuated SD12-F120(GI) strain at high (5,000 PFU) dose (A), medium (500 PFU) dose (B), and low (50 PFU) dose (C) and challenged with the indicated JEV strains. Significant differences between groups were statistically tested using the Kaplan-Meier analysis.

genotype JEV-GI/V strain challenge were 100% (Fig. 5A), 100% (Fig. 5B), and 87.5% (Fig. 5C) in groups vaccinated with high, medium, and low doses, respectively. These data suggested the slightly reduced protective efficacy of SA14-14-2(GIII) vaccine against heterologous genotype JEV-GI/V challenge.

**Partial protective efficacy of JEV-GI/V-E5 against heterologous genotype SD12(GI) and N28(GIII) challenge in mice.** The JEV-GI/V-E5 strain displayed an attenuated virulence phenotype in mice. We therefore tested the protective effect of JEV-GI/V-E5, as a model GV vaccine, against challenge with its virulent parental strain JEV-GI/V and with heterologous genotype strains SD12(GI) and N28(GIII) in mice. Mice were immunized with JEV-GI/V-E5 at high (5,000 PFU), medium (500 PFU), or low (50 PFU) doses, and challenged with the homologous and heterologous genotype viruses. Immunization with JEV-GI/V-E5 at high, medium, and low doses provided complete protection against challenge with the virulent parental strain JEV-GI/V (Fig. 6). However, the protective efficacy against heterologous genotype viruses was reduced. The protection rates against challenge with heterologous genotype strains SD12(GI) and N28(GIII) were 75% and 100% at high dose (Fig. 6A), 75% and 87.5% at medium dose (Fig. 6B), and 62.5% for both strains at a low dose of vaccine (Fig. 6C), respectively. These data suggested that the attenuated JEV-GI/V-E5 strain provides 100% protection against challenge with the virulent parental strain JEV-GI/V, but only partial protection against challenge with heterologous GI and GIII viruses in mice.

**Histopathological lesions and viral loads in brains of vaccinated mice.** To further explore the cross-protection, brain samples were collected from the vaccinated mice at 7 days postchallenge for analysis of histopathological lesions and viral loads. The naive mice challenged with the virulent SD12(GI), N28(GIII) or JEV-GI/V strain developed encephalitis histopathologically characterized by the multifocal lymphohistiocytic

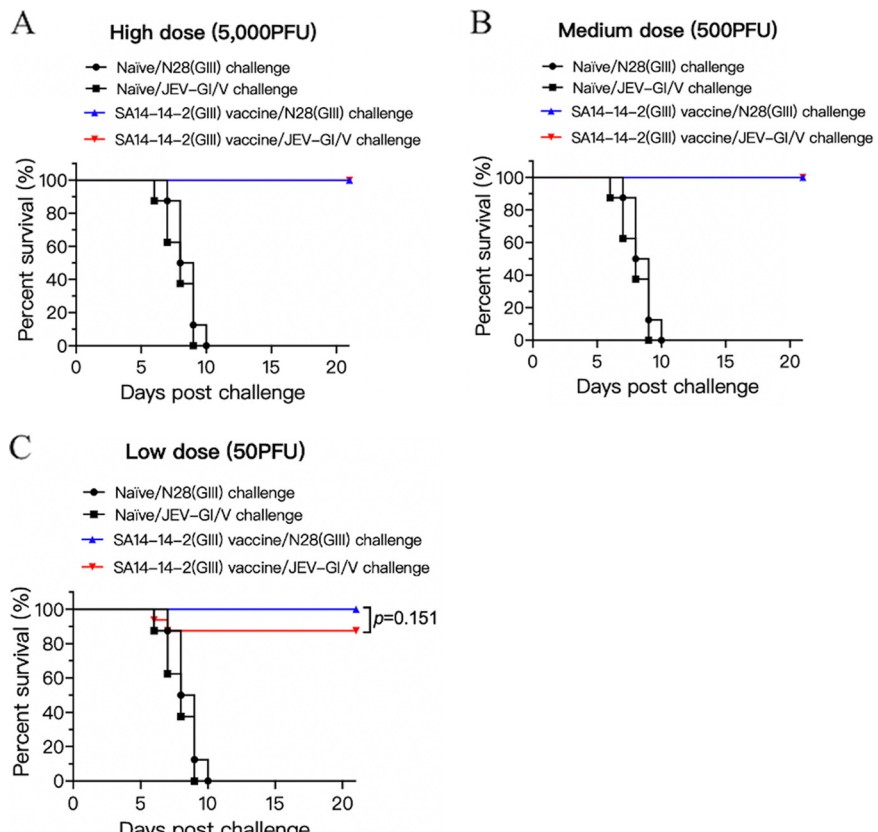

**FIG 5** Protective efficacy of the SA14-14-2 (GIII) vaccine. Mice ($n = 16$) were immunized with the attenuated SA14-14-2(GIII) vaccine at high (5,000 PFU) dose (A), medium (500 PFU) dose (B), and low (50 PFU) dose (C) and challenged with the indicated JEV strains. Significant differences between groups were statistically tested using the Kaplan-Meier analysis.

perivascular cuffs and lymphohistiocytic meningitis (Fig. 7A). JEV NS1 gene expression was detected in the brain samples collected from these mice (Fig. 8A). The histopathological lesions of encephalitis were observed in the vaccinated mice challenged with heterologous genotype viruses, but not in the vaccinated mice challenged with homologous genotype virus (Fig. 7B, C, and D). In addition, the viral NS1 gene expression was detectable in the vaccinated mice challenged with heterologous genotype viruses, but not in the vaccinated mice challenged with homologous genotype virus (Fig. 8B, C, and D). Overall, these data further confirmed the partial cross-protection between different genotype vaccines against heterologous genotype virus challenge.

**Low titer of neutralizing antibodies against heterologous genotype viruses in mice vaccinated with JEV-GI/V-E5.** Neutralizing antibodies play an important role in protection against JEV infection, and the titers of neutralizing antibody correlate with this protective effect (40, 41). Given the partial cross-protection among GI, GIII, and chimeric GI/V strains, we measured the neutralizing antibody titers against the heterologous genotype viruses by the plaque reduction neutralization test (PRNT). Serum samples were collected from vaccinated mice and the $PRNT_{50}$ titers specific to the homologous and heterologous genotype viruses were determined. Serum samples collected from pre-vaccinated mice showed a background level $PRNT_{50}$ titer ($<10$) in all the JEV strains examined (Fig. 9A). Serum samples from mice vaccinated with JEV-GI/V-E5 exhibited a significantly higher $PRNT_{50}$ titer ($50 \pm 10$) against the virulent parental JEV-GI/V strain than against the heterologous genotype SD12(GI) ($12.3 \pm 9.7$) and N28(GIII) ($21.2 \pm 8.8$) strains (Fig. 9B and Table S4). A similar trend was also observed in groups vaccinated with the attenuated SD12-F120(GI) and SA14-14-2(GIII) strains. The $PRNT_{50}$ titer against SD12(GI) in the group vaccinated with SD12-F120(GI) strain was

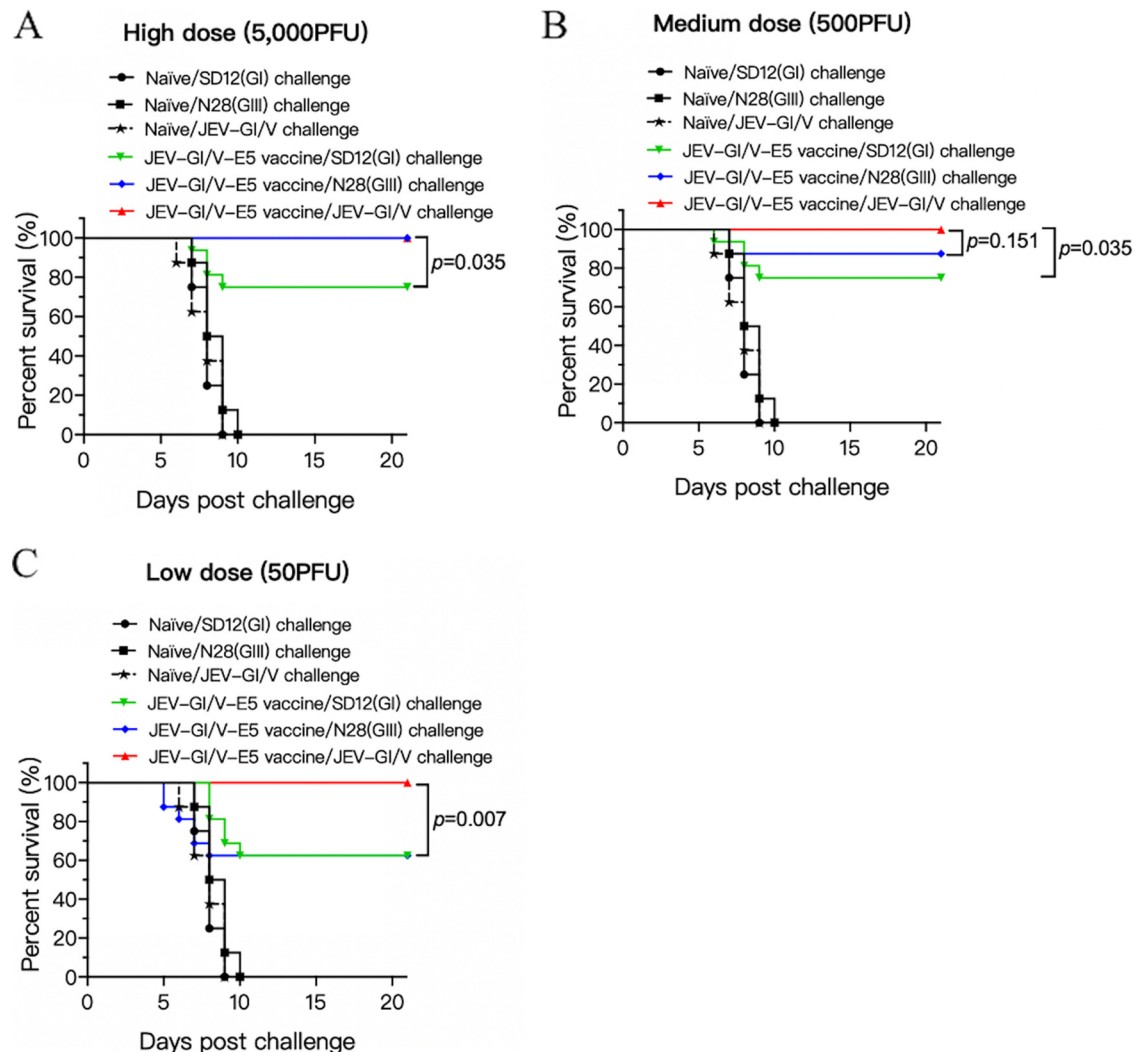

**FIG 6** Protective efficacy of the JEV-GI/V-E5 vaccine. Mice (*n* = 16) were immunized with the attenuated JEV-GI/V-E5 vaccine at high (5,000 PFU) dose (A), medium (500 PFU) dose (B), and low (50 PFU) dose (C) and challenged with the indicated JEV strains. Significant differences between groups were statistically tested using the Kaplan-Meier analysis.

$66.4 \pm 10.4$, which was remarkably higher than that against the heterologous genotype N28(GIII) and JEV-GI/V strains (Fig. 9C and Table S4), while the $PRNT_{50}$ titer against N28(GIII) in the group vaccinated with SA14-14-2(GIII) was $52.2 \pm 9.8$, which was obviously higher than that against the heterologous genotype SD12(GI) and JEV-GI/V strains (Fig. 9D and Table S4). These data indicated the low $PRNT_{50}$ titer of neutralizing antibodies raised against heterologous genotype viruses among different JEV genotypes in mice.

**Amino acid variations in E protein among GI, GIII, and GV strains.** The E protein is associated with virus binding to cellular receptors, membrane fusion, and inducing protective immunity against JEV (29–31). Given the reduced cross-protection among different genotypes, the amino acid variations in E protein were compared among GI, GIII, and GV viruses (Table S5). A BLAST search of the E protein sequence of XZ0934(GV) strain against different genotype viruses revealed 2–6 amino acid variations compared with the homologous genotype Muar (42), K15P38 (43), 10-1827 (44), and other strains (21) (Fig. 10A), respectively. However, 44 to51 and 42 to 48 amino acid variations were observed in E protein between the XZ0934(GV) and GI strains with 89.8% to 91.2% of amino acid homologies, and between the XZ0934(GV) and GIII strains with 90.4% to 91.6% of amino acid homologies, respectively (Fig. 10A). Interestingly, no amino acid variations were observed

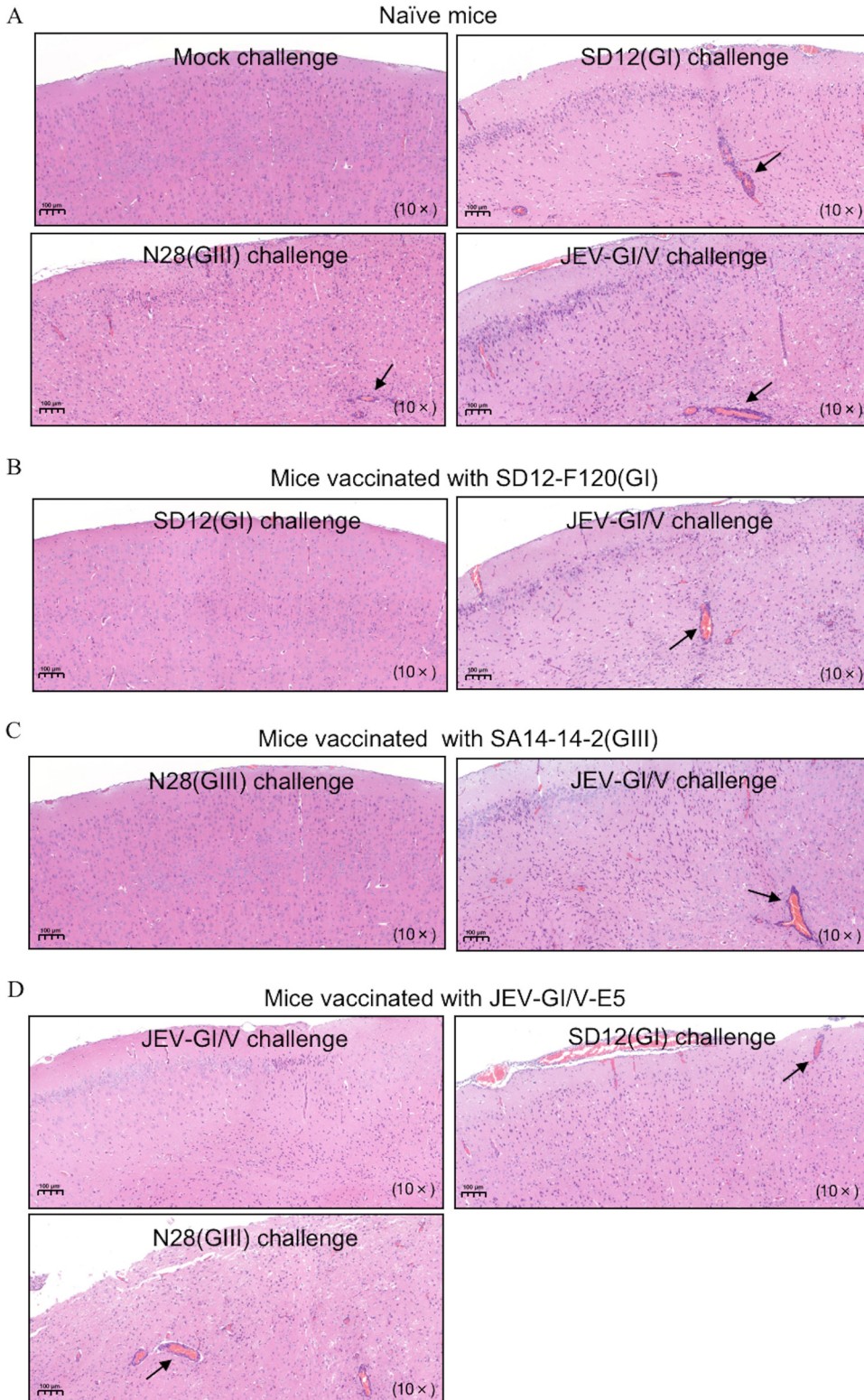

**FIG 7** Histopathological lesions in brains of vaccinated mice. Mice (*n* = 5) were immunized with SD12-F120(GI), SA14-14-2(GIII), or JEV-GI/V-E5 at a dose of 1,000 PFU and challenged with the indicated JEV strains at 14 days postvaccination. Brain samples were collected at 7 days postchallenge for analysis of histopathological lesions. The sectioned brain simples were stained with hematoxylin and eosin. The multifocal lymphohistiocytic perivascular cuffs are indicated by arrows. (A) Histopathological lesions in brain samples collected from naive mice challenged with SD12(GI), N28(GIII) and JEV-GI/V strains. (B) Histopathological lesions in brain samples collected from SD12-F120(GI)-vaccinated mice challenged with SD12(GI) and JEV-GI/V strains. (C) Histopathological lesions in brain samples collected from SA14-14-2(GIII)-vaccinated mice challenged with N28(GIII) and JEV-GI/V strains. (D) Histopathological lesions in brain samples collected from JEV-GI/V-E5-vaccinated mice challenged with SD12(GI), N28(GIII) and JEV-GI/V strains.

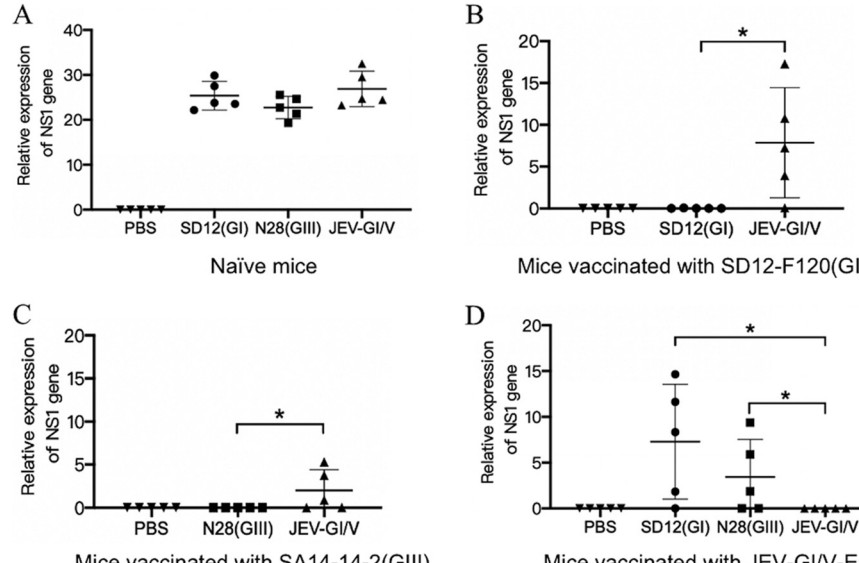

**FIG 8** Viral loads in brains of vaccinated mice. Mice (*n* = 5) were immunized with SD12-F120(GI), SA14-14-2(GIII), or 1JEV-GI/V-E5 at a dose of 1,000 PFU and challenged with the indicated JEV strains at 14 days postvaccination. Mock challenge was performed using PBS. Brain samples were collected at 7 days postchallenge for analysis of viral loads. The viral load was detected by qRT-PCR with primer pair specific to JEV NS1 gene. (A) Relative expression of NS1 gene in brain samples collected from naive mice challenged with SD12(GI), N28(GIII) and JEV-GI/V strains. (B) Relative expression of NS1 gene in brain samples collected from SD12-F120(GI)-vaccinated mice challenged with SD12(GI) and JEV-GI/V strains. (C) Relative expression of NS1 gene in brain samples collected from SA14-14-2(GIII)-vaccinated mice challenged with N28(GIII) and JEV-GI/V strains. (D) Relative expression of NS1 gene in brain samples collected from JEV-GI/V-E5-vaccinated mice challenged with SD12(GI), N28(GIII) and JEV-GI/V strains. Significant differences between groups were analyzed by Student's *t* test. *, $P < 0.05$.

at positions 47, 107, 138, 123, and 176, which are considered to be responsible for JEV virulence (11, 13, 35, 36), among the 3 genotype strains examined. However, 40 amino acid mutations were observed to be unique to GV strains, which were located at different sites throughout the E protein, including the antigenic epitope regions and an immunodominant epitope (45, 46). This implied the potential role of these variations in antibody recognition and immune evasion of GV strains (47, 48). For example, the variations of S64→T64, T66→A66, E83→T83, K84→R/K84, S149→A149, S156→T/S156, V159→I159, G261→A261, A311→S311, T/S327→Q327, S329→T329, S331→T331, M374→L374, Y382→F382, and T402→S402 were present in the antigenic epitope regions, including the region recognized by most of the existing JEV cross-reactive monoclonal antibodies (45) (Fig. 10A and B). JEV E protein is a largely exposed structural protein harboring both B-cell and T-cell epitopes and plays dominant roles, as a major immunogenic antigen, in inducing both humoral and cellular immune response against JEV infection (49). These unique amino acid variations in E protein indicated that the antigenicity of GV strains may be different from other genotypes, thereby likely contributing to the poor cross-protection of JEV-GI/V-E5 against challenge with GI and GIII viruses.

## DISCUSSION

JEV GV has reemerged in China and South Korea (21) after more than 50 years. The newly emerged genotype strain is both genetically and serologically different from other genotypes (26–28, 50). The structural proteins of flavivirus play important roles in the determination of virulence and the induction of an immune response (6–9). This study aimed to determine the role of the structural proteins of GV strain in virulence and cross-protection against different genotypes in mice.

To achieve this, we replaced the structural proteins (C-prM-E) of the attenuated SD12-F120(GI) strain with those of the virulent XZ0934(GV) strain to generate the

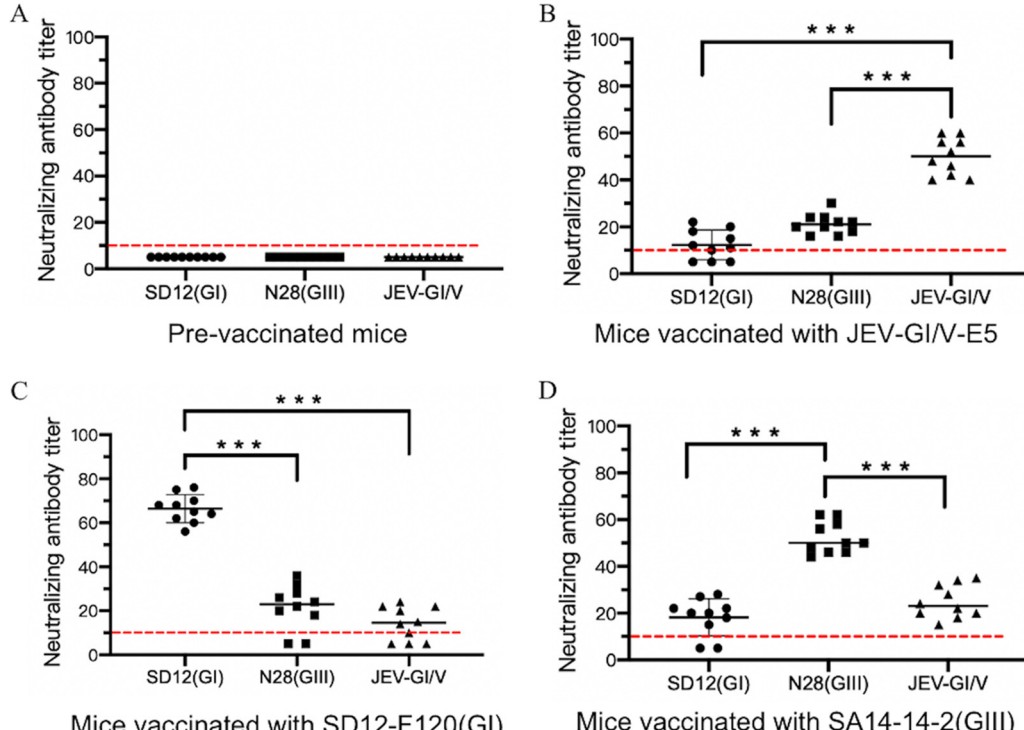

**FIG 9** Neutralizing antibody levels in mice vaccinated with JEV vaccines. Mice ($n = 10$) were immunized with SD12-F120(GI), SA14-14-2(GIII), or JEV-GI/V-E5 at a dose of 5,000 PFU, and serum samples were collected at 14 days postvaccination for the detection of neutralizing antibodies against SD12(GI), N28(GIII), and JEV-GI/V strains. (A) PRNT$_{50}$ titers in serum samples collected from pre-vaccinated mice. (B) PRNT$_{50}$ titers against the indicated JEV strains in mice vaccinated with JEV-GI/V-E5. (C) PRNT$_{50}$ titers against the indicated JEV strains in mice vaccinated with SD12-F120(GI). (D) PRNT$_{50}$ titers against the indicated JEV strains in mice vaccinated with SA14-14-2(GIII). A $P$ value was generated by the Student's $t$ test. ***, $P < 0.001$; **, $P < 0.01$. The positive cutoff value (10) for the PRNT$_{50}$ titer is indicated by a red dotted line.

chimeric JEV-GI/V strain in an attenuated SD12-F120(GI) background. The resulting chimeric JEV-GI/V strain, harboring the structural proteins of the virulent XZ0934(GV) strain, showed reduced replication efficiency compared with its parental SD12-F120(GI) strain in BHK-21 cells. However, JEV-GI/V was highly virulent in mice compared with the attenuated phenotype of its parental SD12-F120(GI) strain. The LD$_{50}$ value for JEV-GI/V at which neuroinvasiveness and neurovirulence were observed in mice was lower than those of the virulent SD12(GI) and N28(GIII) strains, but likely similar to that of the virulent SA14(GIII) strain. Notably, the LD$_{50}$ value for JEV-GI/V correlating with neuroinvasiveness was $10^{-1.5}$, which was lower than that of the XZ0934(GV) strain ($10^{1.07}$) (26). This difference in neuroinvasiveness was attributable to the difference in species and age of the mice used in our study compared with Cao and colleagues (26). We used 3-week-old C57BL/6 mice for analysis of JEV virulence, while Cao and colleagues used 5- to 6-week-old BALB/c mice (26). The C57BL/6 mouse with dark brown coat color that is an inbred strain derived from the C57BL/6 mice is more susceptible to JEV infection than the BALB/c mouse with white coat color that is an outbred strain originating from the New York mice. In addition, the susceptibility of mice to JEV infection is known to be age-related, where a 3-week-old mouse is more susceptible than a 5- to 6-week-old mouse. Overall, our data suggested that the structural proteins were essential for the determination of virulence in the GV strain, which is in agreement with a previous observation that the structural protein region may play a critical role in the neuroinvasiveness of GV strain, as demonstrated by a chimeric JEV S-g5/NS-g3 that contains the structural protein region from GV strain and the nonstructural protein region from GIII strain (28).

It has been established that the virulence of JEV is mainly determined by its structural proteins, especially the E protein that plays a dominant role in JEV virulence (51,

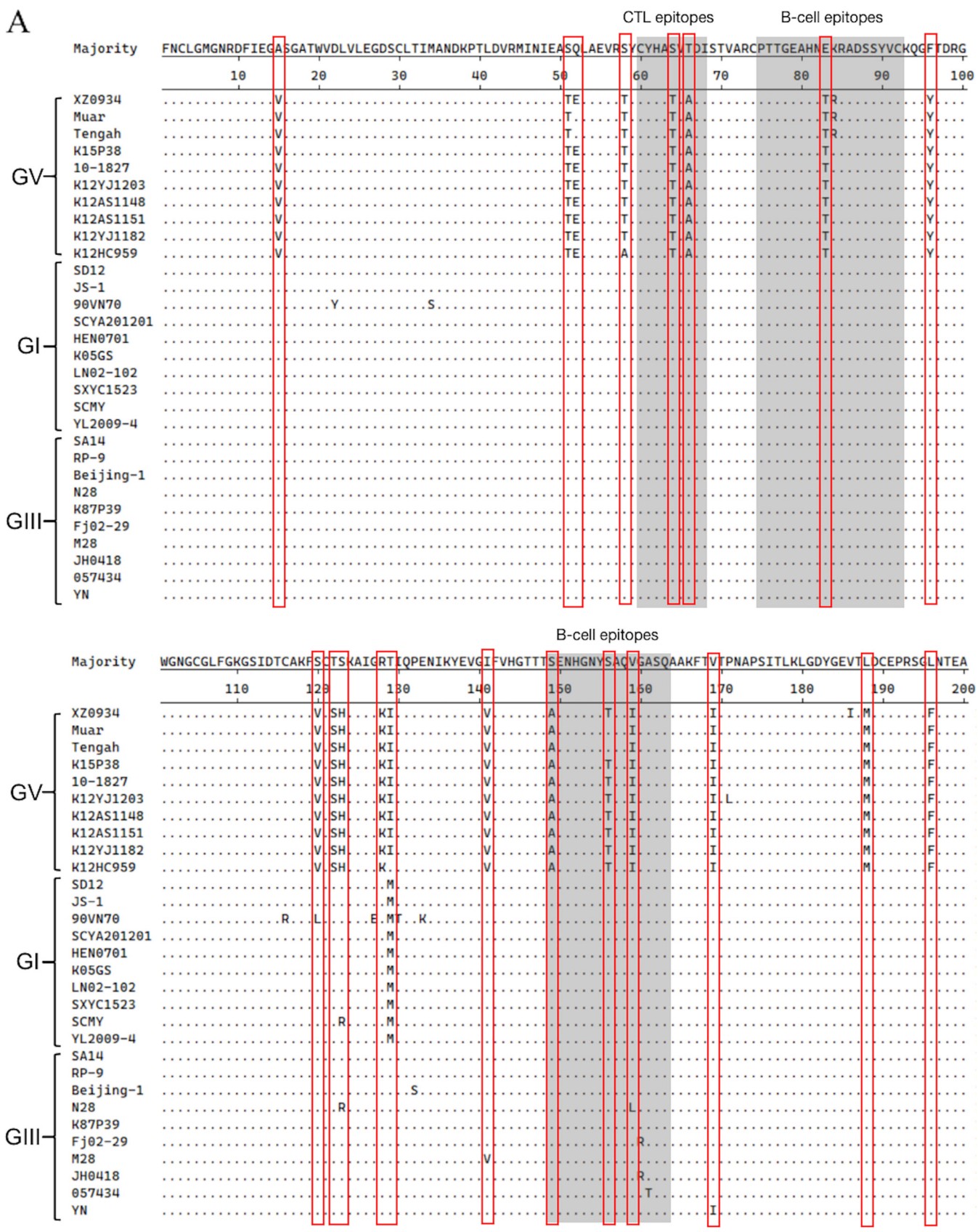

**FIG 10** (Continued)

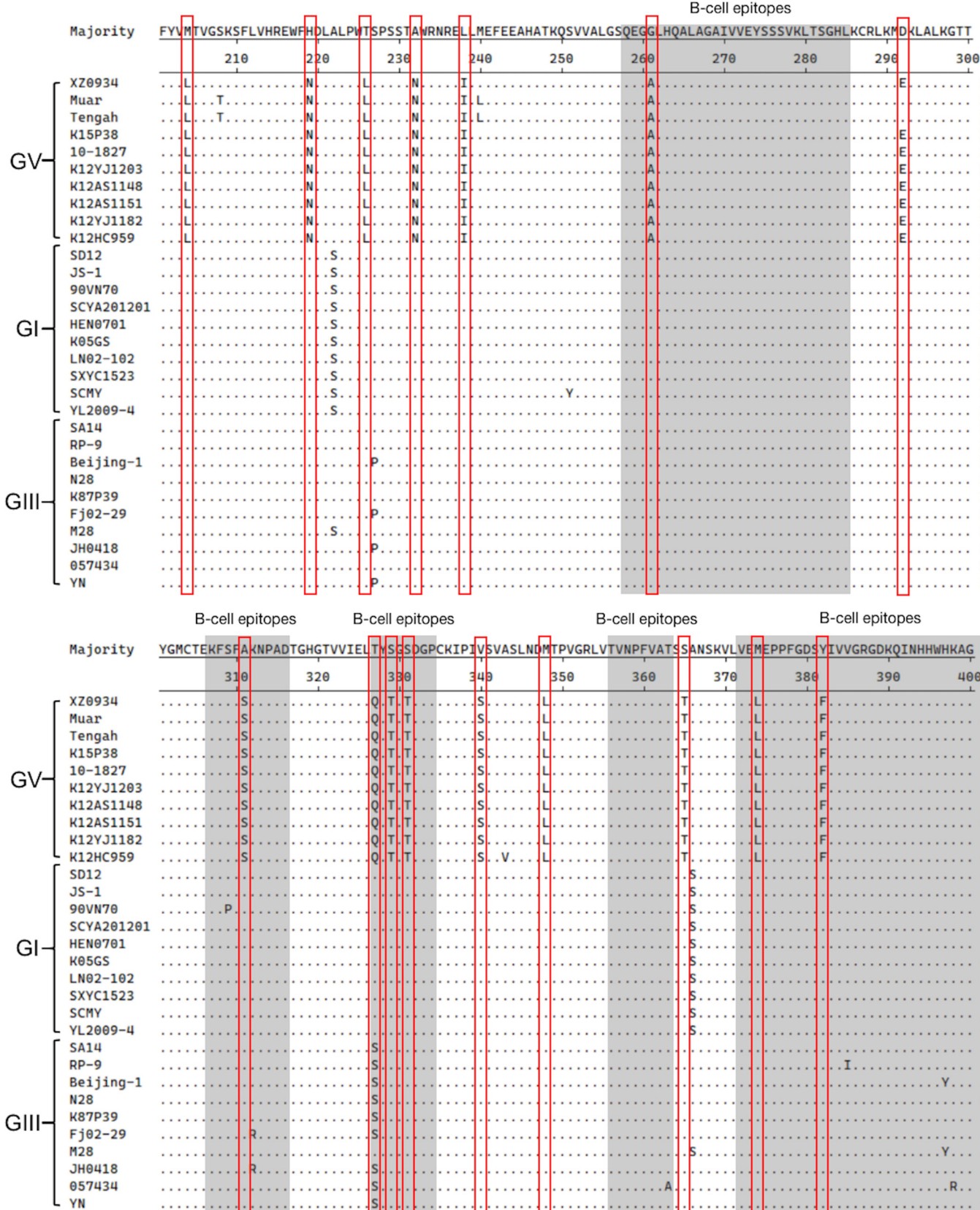

**FIG 10** (Continued)

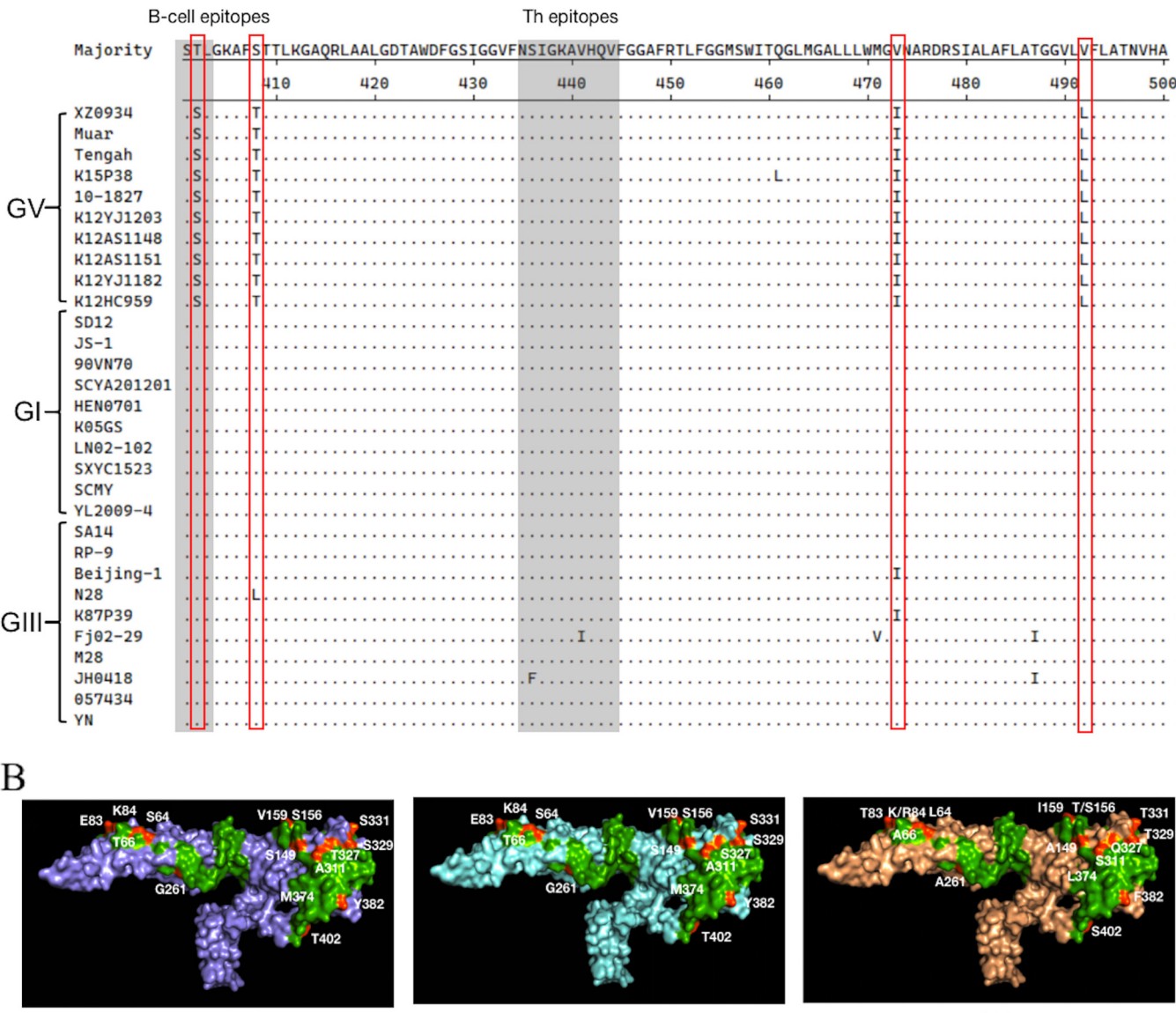

**FIG 10** Amino acid variations in E protein among the GI, GIII, and GV strains. (A) Sequence alignment of E proteins from the GI, GIII, and GV strains. The antigenic epitope regions are shaded in gray. The amino acid variations between GV and other genotypes are boxed in red. (B) The 3D-structure of the homologous E protein dimer. The areas highlighted in green represent the neutralizing epitope regions. The amino acid variations present in the neutralizing epitope regions between GV and other genotypes are highlighted in orange.

52). The E protein is structurally divided into domains I (DI), II, and III (10). The residue at position 138 of E protein is located in the "hinge" region of the interface between DI and DII, and mutation of this position changes the conformation and function of the protein and alters the virulence of GI and GIII strains (13). Residue 47 is connected to residue 138 by a hydrogen bond and contributes to the neurovirulence of the GI strain (13). Residue 107 is located in a highly conserved hairpin motif in DII, and plays a synergistic role with residue 138 in determination of the virulence of GIII strains (11). Residue 123 is located in DII, and mutation alters the critical pH required for conformational change before the fusion of E protein with the endosomal membrane (53). Residue 176, located in DI, also contributes to the neurovirulence of GI strains in mice (35). Taking these previous observations into account, we replaced these particular residues (47, 107, 138, 123, and 176), both individually and in combination, in the E protein of the chemic JEV-GI/V with their counterpart residues in the attenuated strains of other genotypes to examine the role of these residues in the virulence of GV strain.

Intracerebral inoculation of mice with JEV-GI/V mutants harboring the single substitutions (N47K, L107F, E138K, H123R, and I176R) resulted in 100% mortality, whereas mortality was significantly reduced in mice intracerebrally inoculated with JEV-GI/V-E5 harboring a combination of these substitutions. This finding suggested that these residues synergistically, but not individually, play a role in determining the neurovirulence of GV strains. These findings were different from previous observations in GI and GIII strains, in which single mutations of the residues, such as 47, 107, 138, and 176, altered the neurovirulence of GI and GIII strains in mice (11, 13, 35).

Intraperitoneal inoculation of mice with JEV-GI/V mutants containing single substitutions of N47K, L107F, E138K, and H123R, but not I176R, showed reduced mortality, RNAemia and inflammatory cytokine expression compared with JEV-GI/V. In particular, JEV-GI/V-E138K completely lost neuroinvasiveness at doses of $10^3$ and $10^4$ PFU, suggesting the dominant role of residue 138 in determination of the neuroinvasiveness of GV strains. Furthermore, JEV-GI/V-E5 harboring a combination of these substitutions completely lost neuroinvasiveness, suggesting a synergistic effect of these residues on the determination of neuroinvasiveness of GV strains in mice. These findings agreed with previous observations from GI and GIII strains, in which a single mutation of residue 123 (12) and combined mutations of residues 107, 138, and 176 (11) altered the neuroinvasiveness of GI and GIII strains. Interestingly, a single mutation of residue 123 increased the neuroinvasiveness of the GI strain, but did not change the neuroinvasiveness of a chimeric virus rJEV-E$^{Muar-H123S}$-M41 harboring the E protein of GV Muar strain (12). This finding was consistent with our data, which showed that a single mutation of residue 123 had no effect on the mortality of mice inoculated with JEV-GI/V-H123R at the dose used for rJEV-E$^{Muar-H123S}$-M41 (12). These observations, together with the data from the intraperitoneal inoculation in this study, suggested that the determinants responsible for the virulence of GV strain may be different from those identified in the GI and GIII strains. Further mutational analyses are needed, both within the structural protein region and in other regions, to further define the determinants responsible for the virulence of GV strains.

In response to the emergence of the GI strain in most countries in Asia, the cross-protective efficacy of JEV vaccines has been extensively evaluated between GI and GIII strains (25, 54). However, the cross-protective efficacy of JEV vaccines between GV and other genotypes has not been fully evaluated. We observed histopathological lesions of brains, presence of JEV in brains, low titers of neutralizing antibodies, and reduced cross-protective efficacy against JEV-GI/V in mice immunized with GI- and GIII-derived vaccines. These findings agreed with a previous observation that GIII-derived vaccines induce low levels of neutralizing antibodies and provide partial cross-protection against GV strain in mice (26). Using the attenuated JEV-GI/V-E5 strain, we evaluated its protective efficacy against the homologous and heterologous genotype challenge in mice. JEV-GI/V-E5 provided complete protection against challenge with its parental JEV-I/V strain, as demonstrated by survival analysis and the absence of histopathological lesions, and virus in the brains of immunized mice, but reduced cross-protection against challenge with the heterologous genotype SD12(GI) and N28(GIII) strains, in which the histopathological lesions and JEV presence were observed in the brains of immunized mice. Similarly, the titers of neutralizing antibodies induced by JEV-GI/V-E5 against its parental JEV-I/V strain were significantly higher than those against the heterologous genotype SD12(GI) and N28(GIII) strains. Overall, these data demonstrated the reduced cross-protection offered by JEV vaccines among different genotypes, thereby highlighting the need for novel vaccines against GV infection.

The protective efficacy of JEV vaccines is positively associated with the presence of neutralizing antibodies in hosts (55). Neutralizing antibodies are primarily induced by E protein (56) that contains both B and T cell epitopes and contributes to the virus binding to cellular receptors and membrane fusion (49). Previous observations suggested that the poor cross-protective and neutralizing effects among different genotypes may be associated with the low homology of E protein among other genotypes (27, 42, 50).

There are 15 amino acid variations present in the neutralizing epitopes of E protein between GV and other genotypes, which may be responsible for the low cross-neutralizing antibody titers and the reduced cross-protection between GV and other genotypes.

The findings observed in this study were based on the chimeric JEV-GI/V strain containing the structural proteins of GV strain in the attenuated SD12-F120(GI) background. Although our findings regarding the pathogenicity of JEV-GI/V and the reduced cross-protection of JEV-GI/V-E5 against heterologous genotypes in mice were similar to those observed with the chimeric JEV S-g5/NS-g3 strain that contains the structural protein region from GV strain and the nonstructural protein region from GIII strain (28), and with a genotype I-based chimeric rJEV-E$^{XZ/SA14142m}$-M41 strain with attenuation mutations in the E protein of GV strain (57), the data still require further validation. This may be done by, for example, comprehensive analysis using GV strain with mutations in structural proteins, especially E protein, in mice, as well as in a natural host, such as pigs.

We replaced the structural proteins (C-prM-E) of the attenuated SD12-F120(GI) strain with those of the virulent XZ0934(GV) strain to generate the chimeric JEV-GI/V strain in the attenuated SD12-F120(GI) background. The resulting JEV-GI/V strain was highly virulent in mice, showing neuroinvasiveness and neurovirulence. This virulent phenotype was remarkably attenuated by a combination of amino acid substitutions in the E protein, suggesting that the structural proteins were essential for determination of the virulence of GV strain. The attenuated JEV-GI/V-E5 strain provided complete protection and induced high neutralizing antibody titers against its parental JEV-GI/V strain, but only partial cross-protection and low titers of cross-neutralizing antibodies against the heterologous GI and GIII strains in mice.

## MATERIALS AND METHODS

**Ethics statement.** All animal experiments were approved by the Institutional Animal Care and Use Committee of Shanghai Veterinary Research Institute, China (IACUC No: Shvri-mo-2020092006) and performed in compliance with the Guidelines on the Humane Treatment of Laboratory Animals (Ministry of Science and Technology of the People's Republic of China, Policy No. 2006 398).

**Viruses and cells.** The virulent GI strain SD12 (SD12[GI]) (GenBank Accession No. MH753127.1) and GIII strain N28 (N28[GIII]) (GenBank Accession No. MH753126.1) were isolated from pigs (58). The attenuated SD12-F120(GI) strain (GenBank Accession No. MN544779.1) was derived from SD12(GI) following 120 passages in the baby hamster kidney cell line (BHK-21) (14). All JEV strains were grown on BHK-21 cells, which were maintained in Dulbecco's Modified Eagle's Medium (DMEM) (Thermo Fisher Scientific) supplemented with 10% fetal bovine serum (FBS) at 37°C in an atmosphere containing 5% $CO_2$.

**Construction of recombinant chimeric JEV.** The recombinant chimeric JEV was generated as described previously (59). Briefly, the cDNA encoding structural proteins (C-prM/M-E) of the virulent XZ0934(GV) strain (GenBank Accession No. JF915894.1) was chemically synthesized according to the sequence deposited in the GenBank database and inserted into an infectious cDNA clone of the attenuated SD12-F120(GI) strain to replace the structural proteins (Fig. 1A). The resulting infectious cDNA clone was used to generate a full-length viral RNA *in vitro* using an mMessage mMachine T7 kit (Invitrogen). BHK-21 cells were transfected with the resulting viral RNA transcripts using DMRIE-C (Invitrogen) to rescue the recombinant chimeric GI-GV JEV (JEV-GI/V). To generate mutants of JEV-GI/V with amino acid substitutions in E protein, site-directed mutation was employed to substitute the amino acids at positions 47, 107, 123, 138, and 176 of E protein based on the plasmid used for generation of the cDNA clone of JEV-GI/V. The primer sequences used for the construction of recombinant JEV are provided in Table S6. All recombinant viruses rescued from BHK-21 cells were identified by Sanger sequencing to ensure that there were no unwanted mutations present in the viruses.

**Immunofluorescence assay.** BHK-21 cells were infected with JEV at a multiplicity of infection (MOI) of 0.1 and subjected to immunofluorescence assays at 24 hpi. The cells were fixed with 4% paraformaldehyde at room temperature for 30 min and blocked with 5% bovine serum albumin at room temperature for 1 h. After washing with phosphate-buffered saline (PBS), the cells were incubated with anti-NS3 polyclonal antibody (60) at room temperature for 1 h, followed by incubation with Alexa Fluor 488 goat anti-mouse IgG antibody. The stained cells were visualized under a fluorescence microscopy.

**Growth kinetics and viral plaque assays.** BHK-21 cells were precultured in 96-well plates to 80% confluence and infected with JEV at 0.01 MOI. After 1 h of incubation, the inoculants were discarded, and the cells were cultured in DMEM supplemented with 2% FBS. The supernatants of JEV-infected cells were sampled at different time points and titrated with 50% tissue culture infective dose ($TCID_{50}$) assays on BHK-21 cells. For viral plaque assays, BHK-21 cells were infected with 100 PFU of JEV and incubated for 2 h at 37°C. The cells were washed three times with PBS and then cultured in DMEM containing 2%

FBS and 1% agarose for 4 days. The cells were fixed with 4% paraformaldehyde for 2 h at room temperature, and the viral plaques were stained with 0.5% crystal violet.

**Neuroinvasiveness and neurovirulence assays in mice.** Three-week-old weanling female C57BL/6 mice (8 mice/group) (Shanghai SLAC Laboratory Animal Co. Ltd.) were inoculated intraperitoneally (100 $\mu$L/each) or intracerebrally (30 $\mu$L/each) with JEV at doses ranging from $10^0$ to $10^{-3}$ PFU to measure neuroinvasiveness or neurovirulence, respectively. The serially diluted doses of JEV were generated by 10-fold diluting JEV stocks in PBS starting from $10^2$ to $10^{-3}$ PFU. The mice inoculated with JEV were monitored daily for 21 days. Mice showing neurological signs, such as paresis and tremors, were euthanized by $CO_2$ asphyxiation, followed by cervical dislocation according to the Guidelines on the Humane Treatment of Laboratory Animals (Policy No. 2006 398). The 50% lethal dose ($LD_{50}$) values were calculated by the method of Reed and Muench for all viruses (61). Survival curve was plotted using GraphPad prism 8.0. The significant differences in survival rates between groups were tested using Kaplan-Meier analysis.

**Detection of viral RNAemia and cytokine expression in blood of mice.** Three-week-old C57BL/6 mice ($n$ = 4 mice/group) were inoculated intraperitoneally with JEV-GI/V mutants at a dose of $10^3$ PFU (100 $\mu$L/each). Blood samples were collected at 3 dpi for analysis of viral RNAemia and cytokine expression (IL-6, IL-1$\beta$, TNF-$\alpha$, IFN-$\beta$, and IFN-$\gamma$). The levels of RNAemia and cytokine expression were examined by qRT-PCR with specific primers (Table S7).

**Vaccination and challenge.** Three-week-old C57BL/6 mice were randomly divided into the vaccinated and control groups ($n$ = 16 mice/group). Mice were intraperitoneally immunized with GIII-derived SA14-14-2 vaccine (SA14-14-2[GIII]) (Lot. 201706116, Wuhan Keqian Biology), attenuated SD12-F120(GI), or attenuated JEV-GI/V-E5 strains at a dose of 50, 500, or 5,000 PFU per animal and challenged intraperitoneally with a dose of 500 $LD_{50}$ of virulent JEV at 14 days postvaccination. The challenged mice were monitored daily for 21 days. Survival curve was generated using GraphPad Prism 8.0 (GraphPad, La Jolla). The significant differences between groups were analyzed by Kaplan-Meier analysis.

**Analysis of viral loads and histopathological lesions in brain of mice.** Three-week-old C57BL/6 mice were intraperitoneally immunized with SD12-F120 (GI), SA14-14-2(GIII) or JEV-GI/V-E5 strains at a dose of 1,000 PFU per animal, respectively, and challenged intraperitoneally with a dose of 500 $LD_{50}$ of virulent JEV at 14 days postvaccination. Mice were sacrificed after 7 days postchallenge and brain tissues were collected for analysis of viral loads and histopathological lesions. Viral load was detected by qRT-PCR with primer pair specific to JEV NS1 gene (Table S7). For analysis of histopathological lesions, the brain tissues were sectioned and stained with hematoxylin and eosin (H & E, Wuhan Servicebio Technology Co., LTD). Histopathological lesions were assessed by determining the integrated optical density index in 3 fields using Case Viewer software (Wuhan Servicebio Technology Co., LTD) (62).

**Detection of neutralizing antibody titers.** Three-week-old C57BL/6 mice were intraperitoneally inoculated with the SD12-F120(GI), SA14-14-2(GIII), and JEV-GI/V-E5 strains at a dose of 5,000 PFU per animal, respectively. Serum samples were collected at 14 days postinoculation for the detection of neutralizing antibody titers. The titer of neutralizing antibodies in the serum samples collected from the inoculated mice was determined using a plaque reduction neutralization test (PRNT), as described previously (24). Briefly, serum samples were inactivated in a water bath for 30 min at 56℃ and serially diluted. The diluted serum samples were mixed with an equal volume of JEV (200 PFU) and incubated for 1 h at 37℃. The mixture was subsequently dispensed onto BHK-21 cells and incubated for 2 h at 37℃. After 2 h adsorption, the cells were overlaid with 1.2% methylcellulose (Thermo Fisher Scientific) in DMEM containing 2% FBS and incubated for 3–5 days at 37℃. The plaques were stained with crystal violet and counted. Neutralizing antibody ($PRNT_{50}$) titers were calculated as the reciprocal of the highest dilution that reduced the plaque numbers by at least 50% relative to the virus control. The positive cutoff value of the neutralizing antibody titer was defined as $PRNT_{50} \geq 10$ (24, 25).

**Multiple amino acid sequence alignment.** Amino acid sequences of the JEV strains were obtained from the GenBank database (S4 Table). Multiple sequence alignments were performed using DNASTAR Lasergene 7.1 (MegAlign). The phylogenetic tree was generated by the neighbor-joining method using MEGAX. The protein 3D structure was predicted using PyMOL software.

**Statistical analysis.** All data were processed using GraphPad Prism 8.0 or the SPSS software package (version 26.0, SPSS Inc.). The Student's $t$ test or Fisher's exact test were employed for statistical analyses. Kaplan–Meier survival curves were analyzed by the log-rank test for significance. A $P$ value $<$ 0.05 was considered statistically significant.

## SUPPLEMENTAL MATERIAL

Supplemental material is available online only.

**SUPPLEMENTAL FILE 1**, PDF file, 1.2 MB.

## ACKNOWLEDGMENTS

The study was supported by Shanghai Municipal Science and Technology Major Project (No. ZD2021CY001) awarded to Z.M., the Project of Shanghai Science and Technology Commission (No. 22N41900400) awarded to Z.M., the Central Public-interest Scientific Institution Basal Research Fund (No. Y2020PT40) awarded to J.W., the Natural Science Foundation of Shanghai (No. 19ZR1469000) awarded to J.W., and the National Natural Science Foundation of China (No. 32273096) awarded to Z.M.

The funders had no role in the study design, data collection, and analysis, decision to publish, or preparation of the manuscript.

We have no financial conflicts of interest.

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
