## [Reviewer comments · Microbiology Spectrum]

Microbiology Spectrum

Virulence and Cross-protection Conferred by an Attenuated Genotype I-based Chimeric Japanese Encephalitis Virus Strain Harboring the E protein of Genotype V in Mice

Qiqi Xia, Yan Zhang, Yang Yang, Xiaochun Ma, Zhixin Guan, Junjie Zhang, Zongjie Li, Ke Liu, Beibei Li, Donghua Shao, Yafeng Qiu, Jianchao Wei, and Zhiyong Ma

Corresponding Author(s): Jianchao Wei, Shanghai Veterinary Research Institute, Chinese Agricultural Academy of Science

Review Timeline:

Submission Date:	May 30, 2022
Editorial Decision:	July 4, 2022
Revision Received:	September 21, 2022
Accepted:	October 2, 2022

Editor: Holly Ramage

Reviewer(s): Disclosure of reviewer identity is with reference to reviewer comments included in decision letter(s). The following individuals involved in review of your submission have agreed to reveal their identity: Bin Zhou (Reviewer #1)

Transaction Report:

DOI: <https://doi.org/10.1128/spectrum.01990-22>

July 4, 2022

Dr. Jianchao Wei
Shanghai Veterinary Research Institute, Chinese Agricultural Academy of Science
Shanghai
China

Re: Spectrum01990-22 (Virulence and Cross-protection Conferred by an Attenuated Genotype I-based Chimeric Japanese Encephalitis Virus Strain Harboring the E protein of Genotype V in Mice)

Dear Dr. Jianchao Wei:

Thank you for submitting your manuscript to Microbiology Spectrum. Your manuscript has now been evaluated by two reviewers. Both reviewers indicated that this study has scientific merit and that your findings are of significant interest to the broader research community. However, the reviewers have highlighted several points that require modifications to the text and the inclusion of additional data to strengthen your findings. We invite you to submit a revised manuscript that addresses these points.

Link Not Available

Sincerely,

Holly Ramage

Journals Department
Reviewer comments:

Reviewer #1 (Comments for the Author):

In this study, the structural protein (c-prm-e) of GV xz0934 virulent strain was used to replace the structural protein (c-prm-e) of gi-gv virulent strain, and the recombinant gi-gv JEV (jev-gi/v) was constructed to determine the role of these structural proteins in virulence and cross protection. In general, the results of the article are relatively clear, the data are clear and complete, If the

following points can be explained clearly, this study will bring new ideas. The major comments are as follows:

- 1、 A large number of animal experiments have been done in this article, but the determination of various indicators of mice is insufficient. Only the survival status of mice has been recorded. It is suggested to supplement experiments to verify the determination of virulence, such as fluorescence qPCR and immunochemistry.
- 2、 The research on the nerve invasiveness and neurotoxicity of jev-gi/v to mice and the research on jev-gi/ve protein mutants occupy a large amount of space in the article. There is only one table in the results section. It is suggested that the data picture be more intuitive.
- 3、 It is expected that the innate immune response of mice infected with different strains will also be impaired. What are the differences in innate immune responses of mice between different strains? Does the peripheral virus titer of mice infected with virulent strains increase, resulting in more viruses invading the brain?
- 4、 Some statistical analyses were omitted. For example, the authors pointed out in Fig. 3, 4 and 5 that the survival rates of mice infected with different strains were significantly different, but did not show statistical analysis (e.g. Kaplan Meier).
- 5、 In the section of Result, what is the problem that the conclusion of blast about the amino acid variation of E protein of GI, gIII and GV strains wants to explain?
- 6、 Several marks in the figure of the article are not standardized. For example, the vertical coordinates in fig1c and 2C only indicate the unit. It is recommended to replace them with virus title (log₁₀ pfu/ml), add "%" after "percent survival" in Fig3, 4 and 5 vertical coordinates, and Fig6 virus titer has no unit.
- 7、 Please pay attention to the article format. The alignment in the body is not standard. Please check and modify the reference format.

Reviewer #2 (Comments for the Author):

Since the GV JEV showed an increase in epidemic areas, which exhibited higher pathogenicity in mice than the prevalent GI and GIII strains. However, its pathogenicity and the cross-protection offered by JEV vaccines against different genotypes haven't been clarified. In this study, the authors constructed a recombinant chimeric GI-GV JEV (JEV-GI/V) strain to determine the role of the structural proteins in virulence and cross-protection. The results demonstrated that immunization of mice with attenuated strain JEV-GI/V-E5 provided complete protection and induced high neutralizing antibody titers against parental strain JEV-GI/V, but partial cross-protection and low cross-neutralizing antibodies titers against the heterologous GI and GIII strains in mice, suggesting the reduced cross-protection of JEV vaccines among different genotypes.

Major comments:

- 1.Fig. 6: It is not enough to show the PRNT50 titers in serum samples. Viral titers and pathological changes in the brain tissues also need to be measured.
- 2.Fig. 7: Fig.7 cannot reflect the prevalence and variation of JEV strains. Increasing the number of strains of different genotypes is needed.

Minor comments:

- 1.Fig. 1a: Please make drawings to real scale. The lengths of C, prM, E and other non-structural proteins need to match its accurate size.
- Fig. 1d: the Fig. 1d is too blurry and needs a clear picture.
- 2.Fig. 2b: The spacing between the images is not uniform, and the images need to be aligned up and down.
- Fig. 2c: What difference does "*" stand for ? Which time point ? "*" is not clearly clarified.
- 3.Lines 324-328: The LD50 value for JEV-GI/V was lower than that of the XZ0934 (GV) strain. This is unreasonable, how to explain it ?
- 4.Lines 484-487: 'at doses ranging from 100 to 10⁻³ PFU', how to dilute the virus ?
- 5.Lines 507-509 and Table S4: How to calculate PRNT50 ? The detailed steps should be described.

Staff Comments:

Preparing Revision Guidelines

Please return the manuscript within 60 days; if you cannot complete the modification within this time period, please contact me. If you do not wish to modify the manuscript and prefer to submit it to another journal, please notify me of your decision immediately so that the manuscript may be formally withdrawn from consideration by Microbiology Spectrum.

Q1、 Are all the authors' conclusions supported by their data?

Answer: Yes if Revised.

Q2、 Is the manuscript written in standard English and easy to comprehend?

Answer: Yes.

Q3、 Does the study include any large datasets that need to be deposited in a public repository?

If the answer is yes, please use the comment box to give us more details.

Answer: No.

Q4、 Does the work described in this study raise any concerns about biosafety or biosecurity that should be discussed prior to publication by the ASM Responsible Publication Committee?

Answer: No.

Q5、 Have appropriate statistical tests been applied?

Answer: Yes.

Q6、 Comments and Suggestions for the Author:

Since the GV JEV showed an increase in epidemic areas, which exhibited higher pathogenicity in mice than the prevalent GI and GIII strains. However, its pathogenicity and the cross-protection offered by JEV vaccines against different genotypes haven't been clarified. In this study, the authors constructed a recombinant chimeric GI-GV JEV (JEV-GI/V) strain to determine the role of the structural proteins in virulence and cross-protection. The results demonstrated that immunization of mice with attenuated strain JEV-GI/V-E5 provided complete protection and induced high neutralizing antibody titers against parental strain JEV-GI/V, but partial cross-protection and low cross-neutralizing antibodies titers against the heterologous GI and GIII strains in mice, suggesting the reduced cross-protection of JEV vaccines among different genotypes.

Major comments:

1. Fig. 6: It is not enough to show the PRNT₅₀ titers in serum samples. Viral titers and pathological changes in the brain tissues also need to be measured.
2. Fig. 7: Fig.7 cannot reflect the prevalence and variation of JEV strains. Increasing the number of strains of different genotypes is needed.

Minor comments:

1. Fig. 1a: Please make drawings to real scale. The lengths of C, prM, E and other non-structural proteins need to match its accurate size.

Fig. 1d: the Fig. 1d is too blurry and needs a clear picture.

2. Fig. 2b: The spacing between the images is not uniform, and the images need to be aligned up and down.

Fig. 2c: What difference does ‘*’ stand for ? Which time point ? ‘*’ is not clearly clarified.

3. Lines 324-328: The LD₅₀ value for JEV-GI/V was lower than that of the XZ0934 (GV) strain.

This is unreasonable, how to explain it ?

4. Lines 484-487: ‘at doses ranging from 10⁰ to 10⁻³ PFU’, how to dilute the virus ?

5. Lines 507-509 and Table S4: How to calculate PRNT₅₀ ? The detailed steps should be described.

Reviewer #1 (Comments for the Author):

In this study, the structural protein (c-prm-e) of GV xz0934 virulent strain was used to replace the structural protein (c-prm-e) of gi-gv virulent strain, and the recombinant gi-gv JEV (jev-gi/v) was constructed to determine the role of these structural proteins in virulence and cross protection. In general, the results of the article are relatively clear, the data are clear and complete, If the following points can be explained clearly, this study will bring new ideas.

The major comments are as follows:

1、 A large number of animal experiments have been done in this article, but the determination of various indicators of mice is insufficient. Only the survival status of mice has been recorded. It is suggested to supplement experiments to verify the determination of virulence, such as fluorescence qPCR and immunochemistry.

Authors' response: Thank you for constructive comment. We have added the data of RNAemia in blood and viral loads in the brains of mice infected with JEV-GI/V mutants to show the difference in virulence among different mutants in Fig. S1 and S2 of the revision (lines 207-240). In addition, the histopathological lesions and viral loads in vaccinated mouse brains following challenge of homologous and heterologous genotype viruses have been provide in Fig. 7 and 8 of the revision to confirm the partial cross-protection (lines 290-304).

2、 The research on the nerve invasiveness and neurotoxicity of jev-gi/v to mice and the research on jev-gi/v e protein mutants occupy a large amount of space in the article. There is only one table in the results section. It is suggested that the data picture be more intuitive.

Authors' response: Thank you for constructive comment. We replaced the Table with survival curves of mice injected with JEV-GI/V mutants (Fig. 3). The table with mouse mortality has been provided as supplementary data in Table S3.

3、 It is expected that the innate immune response of mice infected with different strains will also be impaired. What are the differences in innate immune responses of mice between different strains? Does the peripheral virus titer of mice infected with virulent strains increase, resulting in more viruses invading the brain?

Authors' response: Thank you for constructive comment. To explore the innate immune response in C57BL/6 mice, we tested the levels of inflammatory cytokines (IL-6, IL-1 β , TNF- α , IFN- β , and IFN- γ) in the blood of mice infected with JEV-GI/V mutants. We found that the levels of inflammatory cytokines were co-related with the virulence of JEV-GI/V mutants. For example, the virulent JEV-GI/V induced the highest levels of inflammatory cytokines, whereas the attenuated JEV-GI/V-E5 induced the lowest levels of inflammatory cytokines among the mutants. These data have been provided in Fig. S1 of the revision. In addition, the levels of RNAemia in blood was co-related with the levels of viral loads in brain and virulence, suggesting that high level of the peripheral virus titer resulted in more viruses invading the brain. These data have been added in Fig. S1 and S2 of the revision.

4. Some statistical analyses were omitted. For example, the authors pointed out in Fig. 3, 4 and 5 that the survival rates of mice infected with different strains were significantly different, but did not show statistical analysis (e.g. Kaplan Meier).

Authors' response: Thank you for constructive comment. The significant differences in survival rates between different groups have been analyzed using the Kaplan-Meier method and provided in the Fig. 3, 4, 5, 6 of the revision.

5. In the section of Result, what is the problem that the conclusion of blast about the amino acid variation of E protein of GI, GIII and GV strains wants to explain?

Authors' response: Thank you for constructive comment. The E protein is associated with virus binding to cellular receptors, membrane fusion, and inducing protective immunity against JEV. We compared the amino acid sequences of E proteins from 30 strains of GI, GIII and GV, analyzed the mutations and the location of the mutation sites, and found many amino acid variations among GV, GI and GIII genotypes, of which 40 amino acid mutations were observed to be unique to GV strains. These mutations were located at different sites throughout the E protein, including the antigenic epitope regions and an immunodominant epitope, implying the potential role of these variations in antibody recognition and immune evasion of GV strains. Based on this analysis, we thought that these unique amino acid variations in E protein of GV strains may play roles in the difference in the antigenicity of GV strains from other genotypes as well as the poor

cross-protection of JEV-GI/V-E5 against challenge with GI and GIII viruses.

6、 Several marks in the figure of the article are not standardized. For example, the vertical coordinates in fig1c and 2C only indicate the unit. It is recommended to replace them with virus title (log10 pfu/ml), add "%" after "percent survival" in Fig3, 4 and 5 vertical coordinates, and Fig6 virus titer has no unit.

Authors' response: Thank you for constructive comment. We have adjusted the ordinates of figures 1C and 2C to "Viral titer (Log10 PFU/mL)" (Figure 1C, Figure 2C). We replaced the Fig. 3, 4, 5 with Fig. 4, 5, 6, and we have adjusted the ordinates of figures 3, 4 and 5 to "Percent survival (%)" (Figure 4, Figure 5, Figure 6). We replaced the Fig. 6 with Fig. 9, we have adjusted the ordinates of figure to "Neutralizing antibody titer" (Figure 9).

7、 Please pay attention to the article format. The alignment in the body is not standard. Please check and modify the reference format.

Authors' response: Thank you for constructive comment. We have made adjustments to the alignment of the body of the article, and checked and revised the bibliography.

Reviewer #2 (Comments for the Author):

Since the GV JEV showed an increase in epidemic areas, which exhibited higher pathogenicity in mice than the prevalent GI and GIII strains. However, its pathogenicity and the cross-protection offered by JEV vaccines against different genotypes haven't been clarified. In this study, the authors constructed a recombinant chimeric GI-GV JEV (JEV-GI/V) strain to determine the role of the structural proteins in virulence and cross-protection. The results demonstrated that immunization of mice with attenuated strain JEV-GI/V-E5 provided complete protection and induced high neutralizing antibody titers against parental strain JEV-GI/V, but partial cross-protection and low cross-neutralizing antibodies titers against the heterologous GI and GIII strains in mice, suggesting the reduced cross-protection of JEV vaccines among different genotypes.

Major comments:

1. Fig. 6: It is not enough to show the PRNT50 titers in serum samples. Viral titers and

pathological changes in the brain tissues also need to be measured.

Authors' response: Thank you for constructive comment. We have added viral loads and histopathological lesions in the brains of vaccinated mice following challenge in Fig. 7 and 8 of the revision to confirm the partial cross-protection according to your comment.

2. Fig. 7: Fig.7 cannot reflect the prevalence and variation of JEV strains. Increasing the number of strains of different genotypes is needed.

Authors' response: Thank you for constructive comment. We have increased the number of GI, GIII and GV JEV strains to 30 according to your comment, and analyzed the amino acid variations of E protein among these strains. The resulting data have been provided in Fig. 10 of the revision.

Minor comments:

1.Fig. 1a: Please make drawings to real scale. The lengths of C, prM, E and other non-structural proteins need to match its accurate size.

Authors' response: Thank you for constructive comment. We have remapped the actual scale of fragment sizes for JEV C, prM, E proteins and nonstructural proteins (Fig. 1A) according to your comment.

Fig. 1d: the Fig. 1d is too blurry and needs a clear picture.

Authors' response: Thank you for constructive comment. We have replaced Figure 1D with a clearer picture (Figure 1D).

2.Fig. 2b: The spacing between the images is not uniform, and the images need to be aligned up and down.

Authors' response: Thank you for constructive comment. We have adjusted the spacing of the images and aligned them up and down (Figure 2B).

Fig. 2c: What difference does '*' stand for ? Which time point ? '*' is not clearly clarified.

Authors' response: Thank you for constructive comment. We have changed the "*" to the expression of $p < 0.05$, which was tested by the Student's t-test at 36-72 h post infection. We have added this sentence to Fig. 2C legend of the revision.

3.Lines 324-328: The LD50 value for JEV-GI/V was lower than that of the XZ0934 (GV) strain. This is unreasonable, how to explain it ?

Authors' response: Thank you for constructive comment. The specific reason is probably attributable to the differences in species and age of mice used. We have added explanation in the Discussion of the revision as following.

This difference in neuroinvasiveness was attributable to the difference in species and age of the mice used in our study compared with Cao and colleagues. We used 3-week-old C57BL/6 mice for analysis of JEV virulence, while Cao and colleagues used 5- to 6-week-old BALB/c mice. The C57BL/6 mouse with dark brown coat color that is inbred strain derived from the C57BL/6 mice is more susceptible to JEV infection than the BALB/c mouse with white coat color that is outbred strain originated from the New York mice. In addition, the susceptibility of mice to JEV infection is known to be age-related, the younger, the more susceptible.

4.Lines 484-487: 'at doses ranging from 100 to 10⁻³ PFU', how to dilute the virus ?

Authors' response: Thank you for constructive comment. We observed that all mice inoculated with $\geq 10^3$ PFU died in our pre-experiment. In order to calculate the LD₅₀ of JEV-GI/V, we ten-fold diluted JEV stocks in PBS starting from 10^2 PFU to generate a series of dilutions from 10^2 to 10^{-3} PFU (10^2 , 10^1 , 10^0 , 10^{-1} , 10^{-2} , 10^{-3}). The method for diluting virus stock has been added in the Materials and methods section of the revision.

5.Lines 507-509 and Table S4: How to calculate PRNT₅₀ ? The detailed steps should be described.

Authors' response: Thank you for constructive comment. We followed the references 24 and 25 (Fan YC., 2012, Wei J., 2019) to calculate PRNT₅₀. We have supplemented the detailed steps for calculating PRNT₅₀ in the Materials and methods section of the revision.

October 2, 2022

Dr. Jianchao Wei
Shanghai Veterinary Research Institute, Chinese Agricultural Academy of Science
Shanghai
China

Re: Spectrum01990-22R1 (Virulence and Cross-protection Conferred by an Attenuated Genotype I-based Chimeric Japanese Encephalitis Virus Strain Harboring the E protein of Genotype V in Mice)

Dear Dr. Jianchao Wei:

Your manuscript has been accepted, and I am forwarding it to the ASM Journals Department for publication. You will be notified when your proofs are ready to be viewed.

Sincerely,

Holly Ramage
Editor, Microbiology Spectrum
